# Dominance in self-compatibility between subgenomes of allopolyploid *Arabidopsis kamchatica* shown by transgenic restoration of self-incompatibility

Chow-Lih Yew[1,2], Takashi Tsuchimatsu [1,2,3], Rie Shimizu-Inatsugi [1,2], Shinsuke Yasuda[4], Masaomi Hatakeyama[1,2,5], Hiroyuki Kakui[1,6,7,8], Takuma Ohta[9], Keita Suwabe[9], Masao Watanabe [10], Seiji Takayama [4,11] & Kentaro K. Shimizu [1,2,6] ✉

The evolutionary transition to self-compatibility facilitates polyploid speciation. In *Arabidopsis* relatives, the self-incompatibility system is characterized by epigenetic dominance modifiers, among which small RNAs suppress the expression of a recessive *SCR/SP11* haplogroup. Although the contribution of dominance to polyploid self-compatibility is speculated, little functional evidence has been reported. Here we employ transgenic techniques to the allotetraploid plant *A. kamchatica*. We find that when the dominant *SCR-B* is repaired by removing a transposable element insertion, self-incompatibility is restored. This suggests that *SCR* was responsible for the evolution of self-compatibility. By contrast, the reconstruction of recessive *SCR-D* cannot restore self-incompatibility. These data indicate that the insertion in *SCR-B* conferred dominant self-compatibility to *A. kamchatica*. Dominant self-compatibility supports the prediction that dominant mutations increasing selfing rate can pass through Haldane's sieve against recessive mutations. The dominance regulation between subgenomes inherited from progenitors contrasts with previous studies on novel epigenetic mutations at polyploidization termed genome shock.

Polyploid species originating from genome duplication are found in animals, fungi, and plants and they are estimated to share approximately 30% of plant species[1-3]. The typical traits of natural and crop polyploid species include large cell size, distinct distribution range with environmental robustness, and self-fertilization[2-4]. A pioneer of the study of polyploidy, Ledyard Stebbins, pointed out the association between polyploidy and self-fertilization[5-7]. Frequent evolutionary transitions from self-incompatibility to self-compatibility of polyploid species have been attributed to the advantage of reproductive assurance in overcoming minority cytotype exclusion and reduced

[1]Department of Evolutionary Biology and Environmental Studies, University of Zurich, 8057 Zurich, Switzerland. [2]Department of Plant and Microbial Biology, University of Zurich, 8008 Zurich, Switzerland. [3]Department of Biological Sciences, University of Tokyo, Tokyo 113-0033, Japan. [4]Graduate School of Biological Sciences, Nara Institute of Science and Technology, Ikoma 630-0192, Japan. [5]Functional Genomics Center Zurich, 8057 Zurich, Switzerland. [6]Kihara Institute for Biological Research, Yokohama City University, Yokohama 244-0813, Japan. [7]Institute for Sustainable Agro-ecosystem Services, Graduate School of Agricultural and Life Sciences, University of Tokyo, Nishitokyo 188-0002, Japan. [8]Graduate School of Agriculture, Kyoto University, Kyoto 606-8502, Japan. [9]Graduate School of Bioresources, Mie University, Tsu 514-0102, Japan. [10]Graduate School of Life Sciences, Tohoku University, Sendai 980-8577, Japan. [11]Department of Applied Biological Chemistry, Graduate School of Agricultural and Life Sciences, University of Tokyo, Tokyo 113-8657, Japan. ✉e-mail: kentaro.shimizu@uzh.ch

inbreeding depression[4,7–15]. However, Stebbins also argued that polyploid species may suffer from a "retarded" phenotypic evolution because of the masking effect[5,6]. Theoretical studies have suggested that additional alleles or duplicated copies may confer a masking effect on recessive and additive mutations because of redundancy, and therefore a mutation in each duplicated gene may need to be accumulated in allopolyploid species for phenotypic evolution such as the evolution of self-compatibility[3,4,16,17]. Dominance relationships can reconcile these two aspects of polyploidy by alleviating the masking effect[2,11,18,19]. Haldane suggested that dominant or partially dominant mutations contribute more to adaptive evolution than recessive mutations, which is referred to as Haldane's sieve[20]. The principle can be valid for both allelic interactions in diploid species and epistatic interactions between homeologous copies in polyploid species because allotetraploid species are effectively fixed heterozygotes of subgenomes derived from different species. Theoretical studies have shown that mutations that increase the selfing rate would be dominant to pass through the Haldane's sieve[21].

The role of epigenetics in polyploid evolution has long been discussed[3,4,22]. In the synthetic allopolyploid plants of many species, genome-wide instability in gene expression and methylation has been reported and termed as "genome shock"[3,4,22,23]. It has been debated whether novel genetic and epigenetic variants contributed to adaptive evolution of polyploids or were deleterious[3,24,25]. Besides genome-wide studies on small RNA (sRNA) of allopolyploid species[26,27], the dominance relationship of self-incompatibility in Brassicaceae regulated by sRNA has also been studied in polyploid species[2,11,19]. Self-incompatibility in Brassicaceae including *Arabidopsis* and *Brassica* is characterized by the dominance relationship in pollen (hereafter, male dominance) through sRNAs that function as "dominance modifiers"[28–32]. In this self-recognition self-incompatibility system, SRK (*S*-locus receptor kinase) on the female stigma and SCR/SP11 (*S*-locus cysteine-rich protein/*S*-locus protein 11) on the pollen bind together when both of them are derived from the same *S*-haplogroup and then inhibit the growth of self-pollen tubes through the downstream signaling pathway[33–35]. A large number of highly divergent *S*-haplogroups of the female and male specificity genes (*SRK* and *SCR/SP11*, respectively) segregate trans-specifically at the *S*-locus[11]. The *S*-haplogroups typically show dominance-recessiveness relationships, i.e., when *Sx* is dominant to *Sy*, only *Sx*-specificity is functional and increases mating success by enabling mating with non-self individuals that possess *Sy* while preventing self-fertilization caused by *Sx*[36–38]. Both male and female dominance (dominance in stigma) are known in Brassicaceae[36–38], however, male dominance is more frequent, i.e., two *S*-haplogroups often show dominance in pollen but no dominance (called co-dominance) in stigma[36]. The molecular mechanism of male dominance has been revealed by transgenic experiments in diploid species[33]. sRNAs from a dominant *S*-haplogroup at the *S*-locus regulate the tissue-specific methylation of the *SCR* gene of a recessive *S*-haplogroup, thereby suppressing its expression and function[28,30–32]. In natural allotetraploid species in Brassicaceae, sequencing and crossing experiments suggested the relevance of dominance in self-compatibility[2,11,19]. In the crop allotetraploid species *Brassica napus*, loss-of-function mutations were detected in the dominant haplotypes of *SRK* or *SCR*[39,40]. In *Arabidopsis suecica*, a loss-of-function mutation in the *SCR-A* and *mir867* sRNA that has a homology to the recessive *SCRO2* were inherited from *A. thaliana*[41]. In *Capsella bursa-pastoris*, the B subgenome has loss-of-function mutations in *SCR* as well as *mirS3* that has a potential binding site in the A subgenome, although its relevance in *SCR* dominance regulation is unclear due to the lack of full-length *SCR* sequence in the A genome[42]. Dominance in a hybrid between different species and in synthetic allopolyploid plants was also studied[43]. Although these data suggest the importance of dominance in the allopolyploid species[2,11,18,19], functional evidence using the transgenic technique has been awaited to test the relevance of dominance.

*Arabidopsis kamchatica* ($2n = 4x = 32$) is a model allopolyploid species with a broad distribution range including locality with heavy metal, and the transgenic technique using floral dip was established[44–47]. It is a natural self-compatible and allotetraploid species derived from multiple individuals of two progenitor species, *A. halleri* and *A. lyrata*[44,48,49]. Both progenitor species are predominantly self-incompatible, although self-compatible individuals of *A. lyrata* have been reported[49–52]. *A. kamchatica* has two *S*-loci derived from each progenitor species and shows disomic inheritance[18]. *S*-haplogroups A, B, and C segregate at the *S*-locus derived from *A. halleri*, and haplogroups D and E from *A. lyrata*[18]. The genotyping of 49 individuals covering most of the distribution range identified the following combinations of *S*-haplogroups: AD (47%, Takashima accession as a representative in this study, abbreviated as Tak), BD (34%, Potter accession, Pot), BE (8%, Okhotsk accession, Okh), individuals with only a single *S*-haplogroup amplified (6%) and CD (5%)[18]. Despite the self-compatibility of *A. kamchatica*, full-length sequences with no obvious gene-disruptive mutations were found in *SRK-A*, *SRK-B*, *SRK-D*, and *SRK-E*[18]. Furthermore, crossing experiments showed functional female self-incompatibility in individuals with full-length *SRK-A*, *SRK-B*, and *SRK-D*[18]. These data showed that these *SRK* haplogroups as well as downstream female signaling genes retained functionality even though all of these individuals are self-compatible[18]. This suggests that the self-compatibility of *A. kamchatica* was caused by gene-disruptive mutations of the male self-incompatibility gene *SCR* or other genes necessary for self-incompatibility in pollen, although further transgenic evidence is required to show which gene was responsible for the evolution of self-compatibility[53]. Thus far, no experimental evidence on the existence of dominance either in pollen or in stigma has been reported among the three major *S*-haplogroup combinations of *A. kamchatica* (AD, BD, BE), although *S*-haplogroup C is shown to be most recessive both in male and female self-incompatibility in diploid species[37]. The crossing experiments described above also showed that *S*-haplogroups B and D exhibit no dominance (that is, co-dominant) in the pistil of *A. kamchatica*[18], although it is possible that male dominance exists between B and D haplogroups because many pairs of *S*-haplogroups showed dominance only in pollen and co-dominance in stigma[36].

Functional experiments such as transgenic technique and genome editing are important to identify the genes responsible for the evolution of self-compatibility because sequence analysis alone may be confounded by mutations that are difficult to be detected such as those in regulatory regions resulting in expression loss and by secondary degrading mutations in other genes in the self-incompatibility system after the loss of self-incompatibility[11]. In *A. thaliana*, the standard accession Col-0 had a gene-disruptive mutation both in *SCR* and *SRK*[54] but several accessions including Wei-1 have a full-length *SRK*[55]. By the transgenic introduction of a repaired *SCR*, *A. thaliana* Wei-1 restored self-incompatibility[53]. Such transgenic restoration by introducing a single gene shows that the mutation in the gene is responsible for self-compatibility and that all other genes for self-incompatibility are functional. The self-incompatibility system of Brassicaceae involves many genes other than *SCR* and *SRK*, and therefore self-compatibility may be attributed to so-called modifier genes such as *M*-locus protein kinase gene that are not linked to *S*-locus[35], or to an unknown locus in the self-compatible populations of *A. lyrata*[50,56]. Because the sRNAs regulating male dominance are located at the *S*-locus[28,30], the self-compatibility mutation must be located at the *S*-locus for dominance to be linked.

In this study, we experimentally examined the male dominance relationships using the self-incompatible diploid species. Based on the sequence analysis of *SCR* and sRNA sequences in *A. kamchatica*, we experimentally restored the *SCR* sequences of *S*-haplogroups

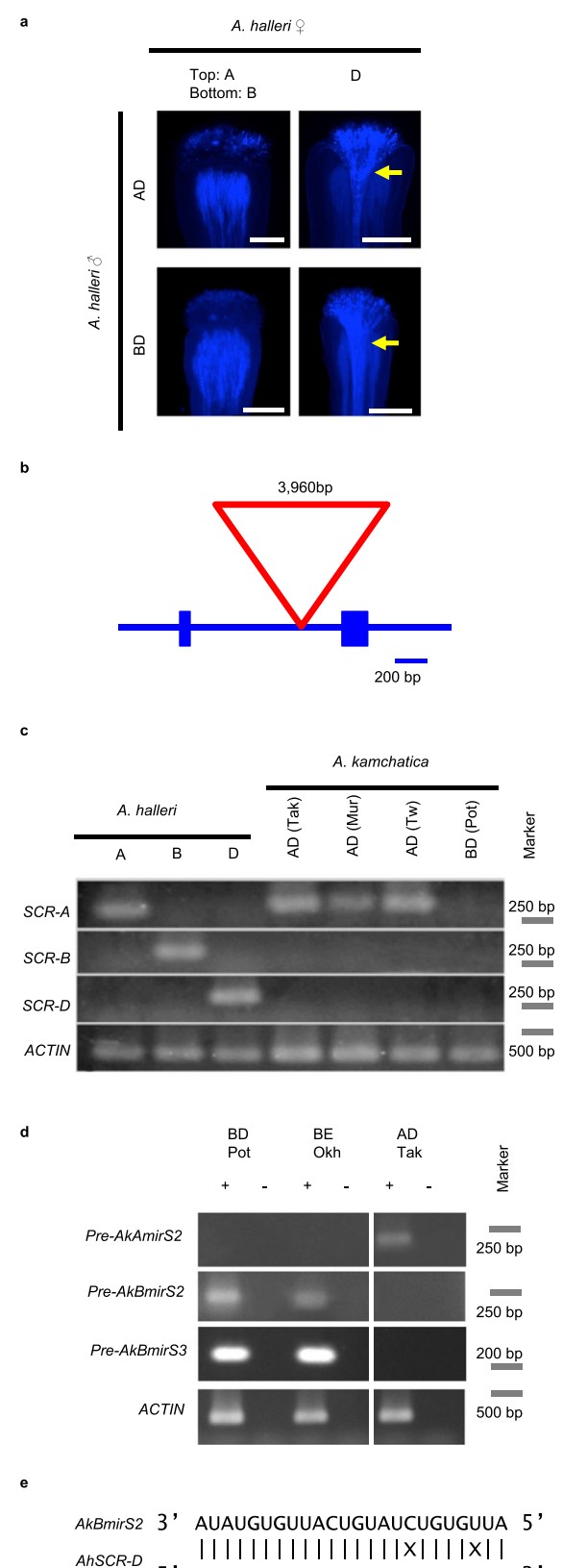

**Fig. 1 | S-haplogroup dominance and expression of SCR genes and small RNAs.**
**a** Representative pictures of the crossings to investigate the dominance hierarchy of the *S*-haplogroups A, B, and D. Stigma of *Arabidopsis halleri* with the haplogroups A (top left), B (bottom left), and D (top and bottom right) crossed with pollen grains derived from heterozygote of AD or BD. Yellow arrows indicate a bundle of pollen tubes. Scale bar = 0.25 mm. At least six observations showed similar results for each crossing (sample numbers in Supplementary Fig. 1a). See Supplementary Table 8 for the genotypes of *A. halleri* individuals. **b** Gene structure of *SCR-B*. Blue boxes: exons; blue lines: intron and intergenic regions; red triangle: insertion in *AkSCR-B*. **c** The expression level of *SCR* genes in the anther cDNA of *A. halleri* and *A. kamchatica* bearing different *S*-haplogroups. *A. kamchatica* accessions from Takashima, Japan (Tak; haplogroups A and D); Murodo, Japan (Mur; haplogroups A and D); Taiwan (Tw; haplogroups A and D); and Potter, Alaska (Pot; haplogroups B and D). *ACTIN* was amplified as a quantitative control. **d** Expression of *AkAmirS2*, *AkBmirS2*, and *AkBmirS3* precursor genes in the anther cDNA of *A. kamchatica* from Potter (Pot; haplogroups B and D), Okhotsk (Okh; haplogroups B and E), and Takashima (Tak, haplogroups A and D). +: Reverse transcription with reverse transcriptase; −: reverse transcription without reverse transcriptase as the negative control because primers amplifying the precursor genes do not flank an intron. **e** Sequence alignment of *AkBmirS2* of dominant *S*-haplogroup B and its target site at the promoter of recessive *SCR-D* obtained from *A. halleri*. *AkBmirS2* sRNA sequences are highly homologous to their target site (22 of 24 nucleotides were identical). X: mismatches.

## Results

### Dominance relationship between *S*-haplogroups in pollen

By exploiting the *trans*-specific sharing of the *S*-haplogroups[18], we examined the dominance relationship and functional sequences using the diploid self-incompatible *A. halleri*. Crossing experiments showed that in the pollen, the *S*-haplogroup B was dominant over D (noted as B > D), that is, the pollen of heterozygote BD was rejected only by the pistil with B specificity, but not by that with D specificity (Fig. 1a, Supplementary Fig. 1a). Similarly, the *S*-haplogroup A was dominant over D (A > D) (Fig. 1a, and Supplementary Fig. 1a). In addition, *S*-haplogroups A and B were co-dominant (A = B) (Supplementary Fig. 1a). Together with a previous report showing *S*-haplogroup D (=Ah12) > E (=Ah02)[37], our data revealed the dominance relationship in pollen, as A = B > D > E. These experiments suggest that *S*-haplogroups A and B are dominant, whereas D and E are recessive in *A. kamchatica*.

### Gene-disruptive mutations in *SCR* genes

The *S*-locus harbors highly divergent sequences of *SRK* and *SCR* and sRNAs[33]. We previously found that female self-incompatibility is functional in some *A. kamchatica* accessions having full-length alleles of *SRK-A, B*, and *D*[18]. Thus, self-compatibility may have evolved through the loss-of-function of *SCR*. We used the transcriptome sequencing approach to isolate *SCR* sequences from *A. kamchatica* and *A. halleri* because *SCR* sequences are too short and highly polymorphic to be isolated using polymerase chain reaction (PCR). Then, we confirmed the linkage to *S*-locus by checking co-segregation with the corresponding *SRK* haplogroup (Supplementary Table 1). The repressed expression in *A. halleri* corresponded to the dominance relationships in pollen as described in the previous section (Supplementary Fig. 1b). We searched for loss-of-function mutations in the dominant *S*-haplogroups of *A. kamchatica*. *AkSCR-B* had a 3,960-bp insertion with homology to a transposable element of the *Mutator* family, *AtMU13*, at the intron region in all tested *A. kamchatica* accessions bearing haplogroup B (Fig. 1b). The expression of *AkSCR-B* was lost in both accessions with the BD and BE haplogroups (Fig. 1c, Supplementary Fig. 1c), strongly suggesting that it is nonfunctional. Among *AkSCR-A* accessions, Taiwan accessions had a frameshift by G to AC substitution, whereas all the other tested accessions had a frameshift by insertion of a 7-bp repeat (Supplementary Fig. 2). Next, we examined the recessive *S*-haplogroups. In agreement with the dominance relationship in *A. halleri*, *AkSCR-E* was not expressed in the BE haplogroup,

B and D by transgenic experiments. Using them, we tested whether the male self-incompatibility specificity gene *SCR* was responsible for the evolution of self-compatibility and whether restoration of the dominant but not the recessive *S*-haplogroup can confer self-incompatibility.

despite the absence of a large indel or obvious loss-of-function mutation (Supplementary Fig. 3). Southern blotting analysis and read mapping of high-throughput genomic sequence data of *A. kamchatica* bearing *S*-haplogroup D to bacterial artificial chromosome (BAC) sequences of *S*-haplogroup Ah12 (=AkD) suggested a deletion in the chromosomal region encompassing *AkSCR-D* (Supplementary Fig. 4), consistent with the lack of *AkSCR-D* expression (Fig. 1c). Due to the lack of *SCR-D* sequence in *A. kamchatica*, that of *A. halleri* is used in the following analyses.

### Isolation of small RNAs and their precursor genes

To examine the involvement of sRNA in *A. kamchatica* self-compatibility, we searched for *S*-located sRNAs that show homology to the genomic region near the two recessive *SCRs*, *SCR-D* and *SCR-E*. First, we found the precursor genes of *mirS2* and *mirS3* sRNAs from each of the two dominant *S*-haplogroups B and A (Supplementary Figs. 5–8, named *AkBmirS2*, *AkBmirS3*, *AkAmirS2*, *AkAmirS3* precursor genes). We confirmed that the former three of them co-segregated with *SCR* sequences at the *S*-locus (Supplementary Table 1). Second, sRNA high-throughput sequencing was conducted to detect sRNA processed from the precursors. Mapping sRNA reads to the *AkBmirS2* precursor gene from the BD (Potter) and BE (Okhotsk) accessions revealed the presence of a 24-nt sRNA in anthers (Supplementary Fig. 6a), and the plant miRNA-target scoring system supported its binding to the promoter of the recessive *SCR-D* (Fig. 1e and Supplementary Fig. 6c). Similarly, mapping sRNA reads to the *AkBmirS3* precursor gene revealed a 24-nt sRNA that targets the intron of *SCR-E* (Supplementary Figs. 7, 8). These findings support the hypothesis that *mirS2* and *mirS3* sRNAs are derived from *S*-haplogroup B precursors and can target *SCR-D* and *SCR-E*, respectively. The 24-nt sRNA of *AkAmirS3* was found in the AD (Takashima) accession (Supplementary Fig. 8b), but not that of *AkAmirS2* (Supplementary Fig. 6b), which was not detected despite the expression of its precursor gene *AkAmirS2* (Fig. 1d). This may indicate a secondary decay of the precursor processing of *AkAmirS2* and provide a unique control material for subsequent experiments.

### Evolutionary restoration of self-incompatibility by repairing a mutation in the dominant *S*-haplogroup

In natural populations, the *S*-haplogroup B was combined with both the recessive haplogroups D and E. The sRNA data suggested that the 3,960-bp insertion in *AkSCR-B* represents a self-compatible mutation that is dominant over the *S*-haplogroups D and E. These data suggested that self-incompatibility should be restored when the function of the dominant *SCR* is restored, but not when that of the recessive *SCR* is restored. Therefore, a restored *AkSCR-B* of Potter accession without the 3960-bp insertion was generated. The restored *AkSCR-B*, under the control of its native promoter, was transformed into Potter accessions (Supplementary Fig. 9a). *AkSCR-B* expression at the anthers was detected in four independent transgenic lines. Self-incompatibility reaction upon self-pollination was detected, except for the line with a low expression (Pot_SCRB_4) (Fig. 2a–f and Supplementary Table 2). Moreover, they produced significantly shorter siliques with fewer seeds per silique than did the wild-type Potter accession (Fig. 2g–i). Further crossing experiments confirmed the viability of the pollen and pistils of these transgenic lines (Supplementary Fig. 9b, c, and Supplementary Tables 3 and 4). These results indicate that self-incompatibility was restored by repairing the single loss-of-function mutation in *AkSCR-B*.

### No restoration of self-incompatibility by repairing a mutation in the recessive *S*-haplogroup

In contrast, four transgenic Potter lines transformed with *AhSCR-D* under the control of its native promoter neither expressed *SCR-D* nor recover self-incompatible reactions upon self-pollination (Fig. 3a, b and Supplementary Table 5). No significant differences in the silique length or number of seeds per silique between the transgenic lines and the wild-type Potter accession were observed (Fig. 3c, d). This result in the recessive *S*-haplogroup contrasts with that in the dominant one; however, it does not rule out the possibility that the *AhSCR-D* genomic fragment is nonfunctional because of the lack of some regulatory elements.

To confirm that the *AhSCR-D* construct is adequate to confer self-incompatibility, it was introduced into the Takashima accession. Consistent with the lack of the 24-nt *mirS2* sRNA in this accession, the expression of *SCR-D* was detected in all seven transgenic lines with a variable expression level. Self-incompatibility reaction was fully or partially observed in all the lines (Supplementary Table 6). Significantly shorter siliques and fewer seeds compared to those in the wild type were observed except for the silique length of the transgenic line Tak_SCRD_9, which showed the weakest *SCR-D* expression (Fig. 4a–d and Supplementary Table 7). Lines with a higher expression level such as Tak_SCRD_8 and Tak_SCRD_13 produced nearly no seeds (Fig. 4a–d). This experiment showed that the *AhSCR-D* genomic fragment is adequate to confer self-incompatibility in this accession. Furthermore, when the *AhSCR-D* promoter was introduced into the self-compatible *A. thaliana*, it was active in the stamen (Supplementary Fig. 10), indicating that the cloned promoter region is sufficient for expression.

## Discussion

In this study, we first found the dominance of *S*-haplogroups A/B over D in pollen, and identified *SCR* sequences and sRNA. Transgenic plants with repaired *SCR* of the dominant *S*-haplogroup B restored the self-incompatibility. Similar to a previous transgenic study of the diploid natural species *A. thaliana*[53], our results indicate that the mutation in the male specificity gene at the *S*-locus was responsible for self-compatibility also in *A. kamchatica*, supporting the adaptive spread of male self-compatible mutations predicted by the theory of sexual asymmetry[12]. Furthermore, our transgenic experiment showed that the self-compatible mutation was located at the *S*-locus, which encompasses the sRNAs responsible for male dominance. By contrast, the self-incompatibility was not restored when the recessive *S*-haplogroup D was repaired in the same accession, although this construct turned out functional in another accession that lacked the *mirS2* sRNA. Together, the data suggest that a single loss-of-function mutation in the self-incompatibility specificity gene *SCR-B* conferred dominant self-compatibility.

The high frequency of loss-of-function of *SCR* or *SRK*, which can be induced by various mutations, such as frameshift and transposable insertion, can explain the prevalence and rapid evolution of self-compatibility in both polyploids and diploids. We suggest that the male dominance relationship plays a role in the evolution of self-compatibility in many natural and domesticated polyploid species[2]. In *Brassica napus*[39,40] or the BE haplogroup of *A. kamchatica*, recessive *S*-haplogroups did not harbor obvious loss-of-function mutations, consistent with the current functional importance of suppression. In *A. suecica*[41], *Capsella bursa-pastoris*[42,43], and individuals with *S*-haplogroups D and C of *A. kamchatica*[18], recessive *S*-haplogroups also had loss-of-function mutations. These mutations may represent secondary decay because self-incompatibility genes are not constrained after becoming self-compatible[53,55]. Alternatively, the mutations may have been segregated or fixed in diploid progenitor species, considering that the transition to self-compatibility is among the most frequent evolutionary transitions in angiosperms[7,11,49,51]. Female dominance can also contribute to the self-compatibility of allopolyploid species. In any case, dominance can allow a single mutation to confer self-compatibility in allopolyploid species, indicating that the existence of duplicated loci did not necessarily "retard" evolution[6]. In addition to duplicated *S*-loci in allopolyploid species, where experimental study is facilitated by fixed heterozygosity, we propose that dominance played a role in heterozygous alleles in diploid species and in each *S*-locus of

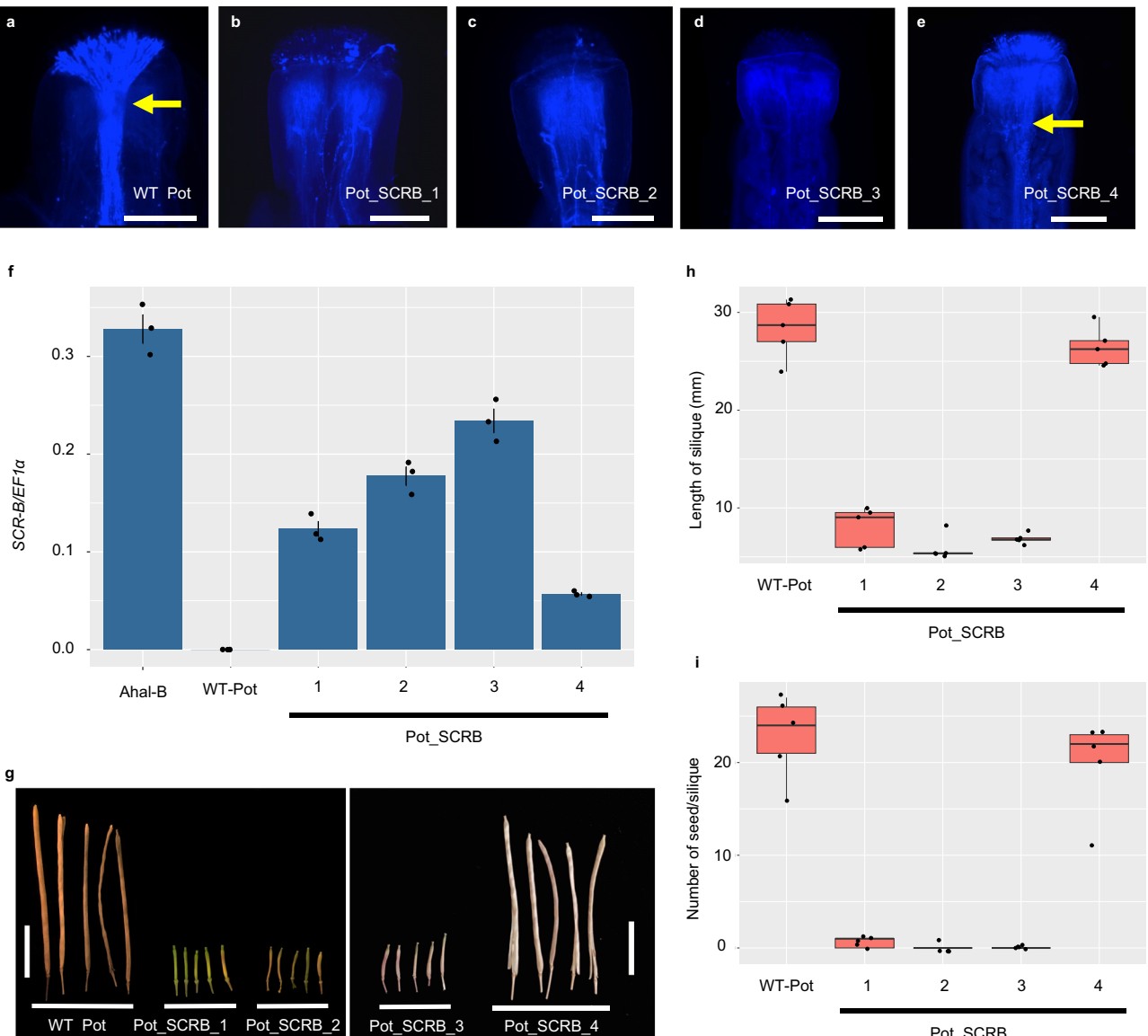

**Fig. 2 | Introduction of restored *AkSCR-B* gene into *Arabidopsis kamchatica* (Potter accession) promoted recovery of self-incompatibility in *A. kamchatica*.** Self-pollination assays of **a** wild-type (WT) Potter, **b** transgenic Potter with restored *AkSCR-B* (Pot_SCRB_1), **c** Pot_SCRB_2, **d** Pot_SCRB_3, and **e** Pot_SCRB_4. At least five observations showed similar results for each (sample numbers in Supplementary Table 2). Yellow arrows indicate a bundle of pollen tubes. Scale bar = 0.25 mm. **f** Expression of *SCR-B* in the anther cDNAs of *A. halleri* bearing haplogroup B (*Ahal-B*), WT Potter, and four transgenic Potter lines (mean ± S.D., $n_{each\ line}$ = 3). **g** Siliques of WT Potter and four transgenic Potter lines. Scale bar = 10 mm. **h** Comparison of silique length from self-pollination of WT Potter and four transgenic Potter lines.

Silique lengths were significantly reduced in the transgenic lines compared to the WT, except for Pot_SCRB_4 (one-way ANOVA and Tukey HSD test, $n_{each\ line}$ = 5). **i** Comparison of the number of seeds per silique from self-pollination of WT Potter and four transgenic Potter lines. The number of seeds per silique was significantly fewer in the transgenic Potter lines than in the WT, except for Pot_SCRB_4 (one-way ANOVA and Tukey HSD test, $n_{each\ line}$ = 5, see Supplementary Table 7 for exact *P*-values). Boxplots **h** and **i** show center line: median; box limits: upper and lower quartiles; whiskers: within 1.5 times the interquartile range; dots: data points. Source data underlying **f**, **h** and **i** are provided as a Source data file.

allotetraploid species transiently during the evolution of self-compatibility until becoming homozygous. The relevance in diploid species is supported by the prevalence of dominant *S*-haplogroups having self-compatible mutations[11]. These findings support the theoretical prediction of Haldane's sieve regarding the prevalence of dominant mutations increasing selfing rate[11,20,21].

Our study highlighted the importance of experimental evidence in addition to sequencing for clarifying complex dominance mechanisms in the Brassicaceae self-incompatibility. The co-dominance of *S*-haplogroups B and D was observed in pistils[18], and a previous report pointed out that *S*-haplogroup D belongs to the most dominant "phylogenetic class"[51] based on phylogenetic analysis of *SRK*

sequences[32]. However, our crossing experiment showed the male dominance of B over D. Furthermore, the evolutionary reversal experiment of *SCR-B* showed that the mutation in *SCR* was responsible for the evolution of self-compatibility, not that in other known or unknown genes such as an *S*-unlinked modifier gene recently reported from a self-compatible population of *A. lyrata*[56]. In this study, we focused on the transgenic experiments of *SCR* but did not manipulate sRNA. The epigenetic regulation of *SCR* expression by sRNA was also supported by transgenic experiments in diploid self-incompatible species[28,30–32], and therefore we also suggest the importance of epigenetic regulation in the dominance of the allopolyploid *A. kamchatica*.

**a**

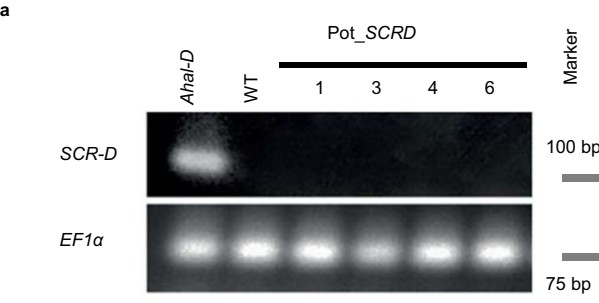

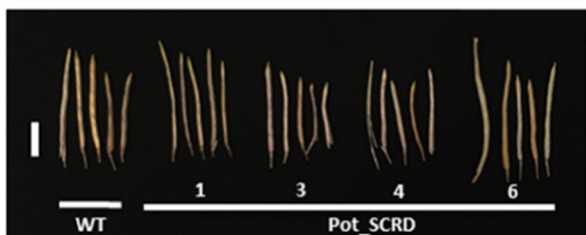

**b**

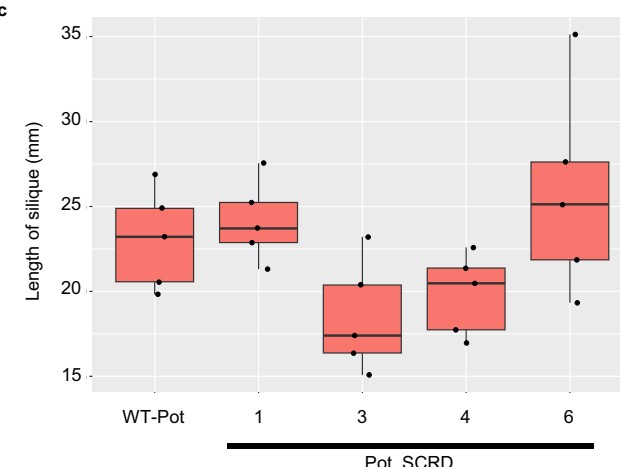

**c**

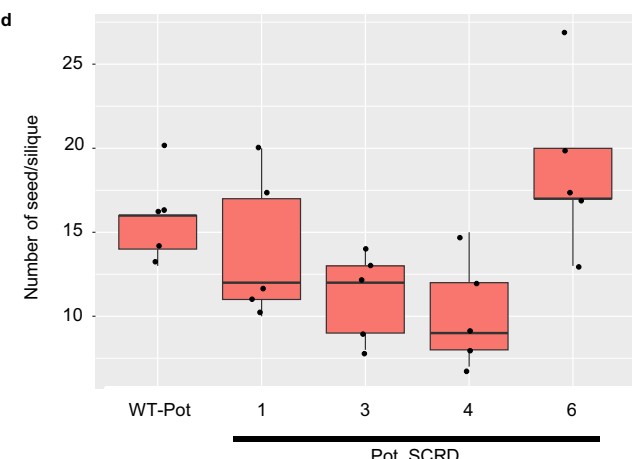

**d**

**Fig. 3 | Self-incompatibility was not restored after the introduction of the repaired recessive *AhSCR-D*. a** *SCR-D* expression in the anther cDNAs of *Arabidopsis halleri* bearing haplogroup D (*Ahal-D*), WT Potter, and four transgenic Potter lines (Pot_SCRD). *ACTIN* was amplified as a quantitative control. **b** Siliques of WT Potter and transgenic Potter lines. Scale bar = 10 mm. **c** Comparison of silique length from self-pollination of WT Potter and transgenic Potter lines. The silique length of transgenic lines was not significantly different from that of the WT. **d** Comparison of the number of seeds per silique from self-pollination of WT Potter and transgenic Potter lines. The length of silique (**c**) or the number of seeds per silique (**d**) in the transgenic lines was not significantly different from that in the WT (one-way ANOVA and Tukey HSD test, $n_{\text{each line}}$ = 5, see Supplementary Table 7 for exact *P*-values). Boxplots **c** and **d** show center line: median; box limits: upper and lower quartiles; whiskers: within 1.5 times the interquartile range; dots: data points. Source data underlying **c** and **d** are provided as a Source data file.

regulation between subgenomes. Many previous studies on the epigenetics of polyploid species focused on the contribution of stochastic epigenetic mutations in synthetic polyploids, which were considered to yield novel variations[3,4,22–25]. Our findings suggest that epigenetic interactions between inherited and combined subgenomes provide a mechanism for facilitating the evolution of a typical trait of polyploidy, i.e., self-compatibility. *Arabidopsis kamchatica* and other polyploid species with transgenic technique[57,58] will be valuable to assess the role of stochastic and inherited epigenetic regulations in other adaptive traits.

## Methods
### Plant materials
Populations of *A. halleri* and *A. kamchatica* used in this study are previously reported[18,48] and listed in Supplementary Table 8. *Arabidopsis kamchatica* (formerly called *Arabidopsis lyrata* subsp. *kamchatica*) is distributed across a broad latitudinal range in East Asia and North America, and can grow in soils with a broad range of the concentration of heavy metals[44,46,47,59]. The plants were grown in a plant chamber under long-day conditions (16 h light/8 h dark cycle, 22 °C). After the production of a few adult leaves, they were moved to a 4 °C chamber (8 h light/16 h dark cycle) for 4–6 weeks for vernalization to induce flowering.

### Isolation of *SCR* genes
Total RNAs were extracted from the anthers of 20 flower buds (stage 12)[60] each of *A. halleri* bearing haplogroups A, B, or D using an RNeasy Plant Mini kit (Qiagen, Hilden, Germany). Complementary DNA (cDNA) was synthesized using the SMARTer kit (TaKaRa Bio USA Inc., San Jose, CA, USA) and submitted for high-throughput sequencing using the Ion Torrent PGM sequencer (Life Technologies, Carlsbad, CA, USA) at the Functional Genomic Center Zurich (Zurich, Switzerland). RNA-seq data were de novo assembled by Trinity[61] (default parameters). A total of 15 known SCR amino acid sequences were used as queries in the TBLASTN homology search against the assembled data as a database (Supplementary Fig. 1d). *AkSCR-E* was isolated using the amino acid sequence of *AhSCR29*[32], which is phylogenetically close to *SCR-E*[18], as a query in the TBLASTN homology searches against the assembled high-throughput genomic sequence data of *A. kamchatica* from Okhotsk that has *S*-haplogroups B and E (assembled by SOAPdenovo2[61] with k-mer = 57 as predicted by KmerGenie[62]). From the TBLASTN results, potential contigs were selected by manually screening for the presence of the eight conserved cysteine residues in SCR. Next, candidates that were detected in more than one *A. halleri* bearing different haplogroups were eliminated because *SCR* genes of different *S*-haplogroups must be highly polymorphic among each other. Primers were designed around the stop codons of the candidates for genotyping by PCR using ExTaq polymerase (TaKaRa Bio Inc., Shiga, Japan) (Supplementary Table 9).

The *S*-locus inherited from *A. halleri* and another inherited from *A. lyrata* were combined in *A. kamchatica*[18]. Our study suggested that the *S*-haplogroup located on one of the two subgenomes was dominant over another *S*-haplogroup located on another subgenome. Analogous to the epigenetic regulation by sRNAs in diploid heterozygotes, our data lead to a hypothesis of epigenetic

**a**

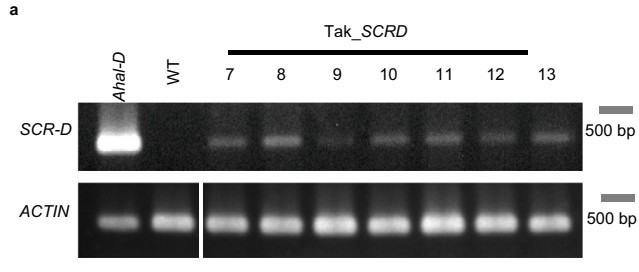

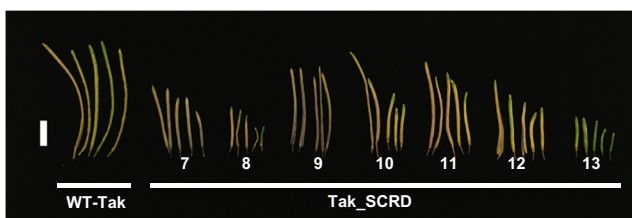

**b**

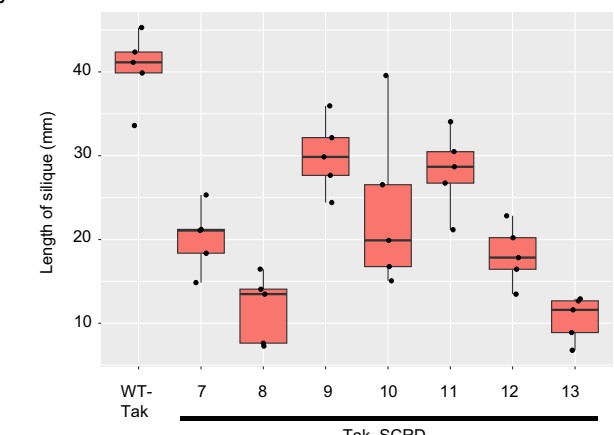

**c**

**d**

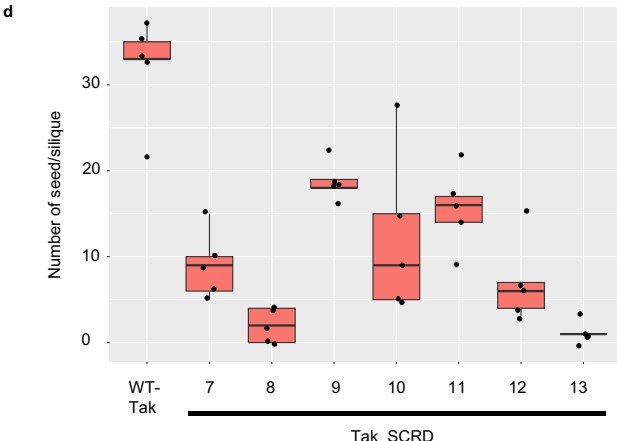

**Fig. 4 | The repaired recessive *AhSCR-D* construct is functional in the Takashima accession, in which *mirS2* small RNA expression is not detectable. a** Expression of *SCR-D* in the anther cDNAs of *Arabidopsis halleri* bearing haplogroup D (*Ahal-D*), WT Takashima accession (WT), and seven transgenic Takashima lines (Tak_SCRD). *ACTIN* was amplified as a quantitative control. **b** Siliques derived from the WT Takashima accession and transgenic lines. Scale bar = 10 mm. **c** Comparison of silique length from self-pollination of WT and transgenic lines. Silique lengths were significantly reduced in the transgenic lines compared with those in the WT, except for in Tak_SCRD_9 (one-way ANOVA and Tukey HSD test, $n_{each\ line}$ = 5).
**d** Comparison of the number of seeds per silique from self-pollination of WT and transgenic lines. The number of seeds per silique was significantly fewer in the transgenic lines than in the WT (one-way ANOVA and Tukey HSD test, $n_{each\ line}$ = 5, see Supplementary Table 7 for exact *P*-values). Boxplots **c** and **d** show center line: median; box limits: upper and lower quartiles; whiskers: within 1.5 times the interquartile range; dots: data points. Source data underlying c and d are provided as a Source data file.

## Isolation of full-length *SCR* genes

Genomic DNA was extracted from the leaf tissues of *A. halleri* and *A. kamchatica* using a DNeasy Plant Mini Kit (Qiagen). Primers were designed using *SCR* genes based on the assembled RNA-seq data to isolate full-length *SCR* genes of *A. halleri* and *A. kamchatica* (Supplementary Tables 8 and 9). Mutations in the *AkSCR* genes were detected by comparing the *SCR* sequences of *A. halleri* and *A. kamchatica* in Geneious 6.1.7[63]. Insertion in *AkSCR-B* was isolated following the instructions of the GenomeWalker™ Universal Kit User Manual (TaKaRa Bio USA Inc., San Jose, CA, USA). The promoter regions of *SCR* genes were also isolated using the same method. DNA samples of *A. halleri* from various populations were genotyped to identify the *A. halleri* bearing haplogroup E for the isolation of full-length *AhSCR-E*.

## Expression of *SCR* genes

Total RNA was extracted from the anthers at stage 12 of *A. halleri* and wild and transgenic *A. kamchatica* using an RNeasy Plant Mini kit (Qiagen). cDNAs were synthesized from 500 ng of each of the total RNA using a High-Capacity RNA-to-cDNA™ Kit (Applied Biosystems, Foster City, CA, USA). Primers flanking the intron of *SCR* genes were used for reverse transcription polymerase chain reaction (RT-PCR) and a reverse transcription quantitative PCR (RT-qPCR) (Supplementary Table 9). The latter was performed using PowerUp SYBR Green Master Mix (A25742) in StepOnePlus Real-Time PCR System of Applied Biosystems. Gel pictures were produced using Gel Doc XR+ Gel Documentation System with Image Lab Software ver. 4.1 (Bio-Rad, Hercules, CA, USA) and ImageJ ver. 2.15.0. *ACTIN* or *EF1α* (*ELONGATION FACTOR 1α*) was amplified as the quantitative control.

## Probe preparation for southern blot

A 317-bp probe designed at the second exon and 3' UTR of *SCR-D* was cloned into the pCR™4-TOPO® vector using the TOPO® TA Cloning® kit (Thermo Fisher Scientific, Waltham, MA, USA) (Supplementary Table 9). Using the vector as a template, the probe was labeled with DIG-11-dUTP via PCR according to the manufacturer's instruction of the PCR DIG Probe Synthesis Kit (Roche Applied Science, Indianapolis, IN, USA). Labeling was confirmed using gel electrophoresis, as the labeled probes have higher molecular weight than unlabeled probes.

## Southern blotting analysis

Genomic DNA was extracted from the leaf tissues of *A. halleri* bearing haplogroup D and three *A. kamchatica* accessions–Murodo, Takashima, and Potter accessions using DNeasy Plant Maxi Kit (Qiagen, Hilden, Germany). A total of 10 μg of each of the accessions was digested with *Xba*I (New England Biolabs, Beverly, MA, USA) at 37 °C overnight. Digested DNA, together with DIG-labeled DNA Molecular Weight Marker VII (Roche Applied Science), was separated using 1% agarose gel (at 100 V for 1 h). Blotting analysis and hybridization were

*SCR* co-segregates with *SRK* of the same haplogroup, as they are tightly linked at the *S*-locus. Using the specific primers that amplify putative *SCR-A*, *SCR-B*, *SCR-E*, *AkSRK-A*, *AkSRK-B*, and *AkSRK-E* genes, segregation of *SCR-A*, *SCR-B*, and *SCR-E* were checked in an F$_2$ population (94 individuals) generated by crossing *A. kamchatica* bearing *S*-haplogroups A and D and *A. kamchatica* bearing *S*-haplogroups B and E (Supplementary Table 1). Additionally, *SCR-D* isolated from *A. halleri* could not be amplified in *A. kamchatica*. Therefore, the segregation pattern of *SCR-D* was examined in F$_2$ populations (32 individuals) generated by crossing the *A. halleri* bearing haplogroup D with *A. halleri* bearing haplogroup B or A (Supplementary Table 10).

performed according to the DIG Application Manual for Filter Hybridization (Roche Applied Science). Capillary transfer of DNA to a positively charged nylon membrane (Roche Applied Science) was performed overnight. DNA was then fixed to the blot via UV cross-linking. The blots were hybridized at 40 °C in 10 ml of hybridization buffer at a concentration of 6 μl of the labeled probes per ml of hybridization buffer. Membranes were washed and chromogenic detections were performed using anti-DIG-alkaline phosphatase and NBT/BCIP (DIG-High Prime DNA Labeling and Detection Starter Kit I, Roche Applied Science) according to the manufacturer's instructions.

### Dominance hierarchy of S-haplogroups A, B, and D

*A. halleri* individuals bearing haplogroup A, B, or D were crossed among each other to generate *A. halleri* bearing haplogroups A and B (AB), *A. halleri* bearing haplogroups A and D (AD), and *A. halleri* bearing haplogroups B and D (BD). To test the dominance relationship between A and D in pollen, the pollen grains of *A. halleri* individuals with AD were applied to the stigmas of *A. halleri* bearing haplogroup A and that bearing haplogroup D. Similarly, to test the dominance relationship between B and D in pollen, the pollen grains of *A. halleri* individuals with BD were crossed to stigmas bearing haplogroup B and those bearing haplogroup D. To test the dominance relationship between A and B, in addition to stigmas, the pollen grains of *A. halleri* individuals with AB were crossed to the stigmas bearing haplogroup A, B, as well as D, the last of which was performed to show that the pollen grains were viable in this co-dominant case. Pollen tube growth was examined using aniline blue staining[18]. S-specificity of *A. halleri* used in this experiment was previously confirmed via interspecific crossings with *A. kamchatica*[18]. The expression of *SCR-A*, *SCR-B*, and *SCR-D* were also examined using these genotypes.

### Isolation of sRNA precursors, detection of sRNA, and mapping

The sRNA precursor of *A. lyrata* bearing haplogroup *Sb* has been previously isolated[31]. Primers were designed based on this precursor to isolate sRNA precursors in *A. kamchatica* (Supplementary Table 9). Both the precursor sequences amplified from *A. kamchatica* bearing haplogroup A and *A. kamchatica* bearing haplogroup B showed high sequence homology with the *mirS2* family, thus designated as *AkAmirS2* and *AkBmirS2*, respectively. The segregation patterns of *AkAmirS2* and *AkBmirS2* were examined in an $F_2$ population (94 individuals) generated by crossing *A. kamchatica* bearing haplogroups A and D and *A. kamchatica* bearing haplogroups B and E. *AkAmirS2* and *AkBmirS2* were perfectly linked to S-haplogroups A and B, respectively (Supplementary Table 1). The secondary structures of the precursor genes were predicted by Geneious 6.1.7[63].

Total RNA was extracted from the anthers of 10 flower buds (stages 9–11) from each of these accessions: *A. kamchatica* from Potter and Okhotsk and Takashima using a mirVana™ miRNA Isolation Kit (Ambion, Austin, TX, USA). To check the expression of sRNA precursors, total RNA was treated with DNase (DNA-free™ DNA Removal Kit, Ambion) and reverse-transcribed to cDNA by High-Capacity RNA-to-cDNA™ Kit (Applied Biosystems). Primers amplifying *AkAmirS2* and *AkBmirS2* were used for RT-PCR (Supplementary Table 9). *ACTIN* was amplified as the quantitative control.

Total RNA before DNase treatment was submitted for small RNA sequencing by Illumina HiSeq 2500 (Illumina, San Diego, CA, USA) at the Functional Genomics Center Zürich. Adaptors were trimmed, and reads shorter than 20 bp were filtered out before being mapped to sRNA precursors to examine the presence of *AkAmirS2* and *AkBmirS2*. Reads were mapped to sRNA precursors using Bowtie 2.3.2 (command: bowtie2 --no-unal -x [file name of reference fasta] -U [file name of fastq])[64] and viewed in the Integrative Genomics Viewer (IGV) 2.3.35[65]. To isolate the sRNA targeting *SCR-E*, known sRNA precursor genes (*mirS1, mirS2, mirS4, mirS5, mir867, mir1887*, and *mir4239*)[32] were used as query in a homology search against the de novo assembled

sequence data of Okhotsk and Takashima, assembled by SOAPdenovo2[66] with k-mer value of 85 as predicted by KmerGenie[62]. High-throughput sRNA reads of Okhotsk anthers were mapped to the potential precursors to detect the presence of 24-nt sRNA using Bowtie 2.3.2[64] and viewed in the IGV 2.3.35[65] as described above. A 24-nt sRNA showing high sequence homology, with a target site at the intron region of *AkSCR-E*, was identified from the *mirS3* precursor gene, and thus designated as *AkBmirS3*. *AkBmirS3* co-segregates with *SCR-B* (Supplementary Tables 1 and 9). The presence of 24-nt *mirS3* targeting *AkSCR-E* was also detected in Takashima, thus designated as *AkAmirS3*. BWA 0.7.12 (command: bwa mem -t8 [file name of reference fasta] [file name of fastq])[67] and IGV 2.3.35[65] were similarly used for the mapping and the visualization of genomic sequences (SAMD00045705, SAMD00089932, SAMD00089933)[49,68] on the BAC sequences of the S-locus region of S-haplogroup Ah12 (Genbank KJ772374.1)[32].

### Cloning and transformation of restored AkSCR-B and AhSCR-D

The full-length sequences, including the promoter and coding sequences of *AkSCR-B* of Potter accession, were restored via a series of PCR amplification using high-fidelity PrimeSTAR GXL DNA polymerase (TaKaRa Bio Inc.) (Supplementary Table 9). First, two different fragments eliminating the insertion were amplified (*AkSCR-B_1°*). Then, 100× diluted PCR products (*AkSCR-B_1°*) were used as the templates for the second round of PCR to amplify the full-length restored *AkSCR-B* (*AkSCR-B_2°*). Since *SCR-D* has been deleted in *A. kamchatica*, *SCR-D* of *A. halleri* was cloned for transformation into *A. kamchatica*. A 3092-bp region of *AhSCR-D*, including 648-bp of promoter regions, two exons, an intron, and 85-bp after stop codon, was amplified by high-fidelity PrimeSTAR GXL DNA polymerase (TaKaRa Bio Inc.) (Supplementary Table 9). Then, 3′ A-overhangs were added to the restored sequences and cloned into pCR®8/GW/TOPO® (Thermo Fisher Scientific) according to the manufacturer's instructions. The sequences were inserted into the destination vector pFAST-G01[69] via LR recombination using LR Clonase® II Enzyme mix (Thermo Fisher Scientific). The vectors were then transformed into *Agrobacterium tumefaciens* GV3101 via electroporation. Constructs generated were transformed into the Potter accession bearing haplogroups B and D and the Takashima accession bearing haplogroups A and D using *Agrobacterium*-mediated floral dip transformation[45].

### Genotyping and phenotyping of transformed A. kamchatica

Transformed seeds were selected by observation under a fluorescence stereomicroscope with a GFP filter (Olympus SZX12, Japan). The DNA was extracted from the leaf tissues of $T_1$ plants, and insertion of generated constructs was confirmed via PCR using primers designed from the vectors and respective inserted *SCR* genes (Supplementary Table 9). Pollination assays were performed to test whether the male components of the transformed *A. kamchatica* are functional. The anthers of transgenic lines were removed from the flower buds at early stage 13 (before anthesis) and visualized under a stereomicroscope to ensure that there is no pollen on the stigmas. They were then manually pollinated with self-pollen. Stigmas were harvested 24 h after pollination, and pollen tube growth was examined using aniline blue staining[18]. Silique length and the number of seeds per silique of the wild type (WT) and transgenic plants were measured. Statistical significance was assessed by the Bonferroni-corrected Mann–Whitney $U$ test and one-way ANOVA and Tukey HSD test in R. In addition to self-pollination, crosses between transformed pollen and WT pistil and between WT pollen and transformed pistil were checked to ensure the viability of the pollen and pistil, respectively.

### Cloning of the AhSCR-D promoter and GUS staining

The 648-bp of the promoter region of *AhSCR-D* was amplified and cloned into the pCR™8/GW/TOPO® vector using a TA Cloning Kit (Thermo Fisher Scientific) according to the manufacturer's

instructions. The promoter was inserted upstream of the β-Glucuronidase (GUS) reporter gene in the binary vector pGWB3 by LR recombination using LR Clonase® II Enzyme mix (Thermo Fisher Scientific). The vectors were then transformed into *Agrobacterium tumefaciens* GV3101 via electroporation. The construct was introduced into *A. thaliana* using the floral dip transformation method. Transformants were selected on 50 μl/ml kanamycin selection plates. Flower buds were collected and immersed in ice-chilled 90% acetone for 20 min. Samples were vacuum-filtrated for 15 min with GUS staining solution containing 1 mM X-Gluc, 50 mM sodium phosphate buffer, 0.1% Triton X-100, 2 mM potassium ferrocyanide, and 2 mM potassium ferricyanide. Samples were incubated overnight at 37 °C and treated with Herr clearing solution before observation under microscope[70].

### Statistics and reproducibility
Statistical analyses were conducted using R (4.2.2). Exact *P*-values are shown in Supplementary Table 7. The results of all RT-PCR gel images were verified by two more independent experiments.

### Reporting summary
Further information on research design is available in the Nature Portfolio Reporting Summary linked to this article.

## Data availability
The sequence data are available in the INSDC databases under accession code, BioProject: PRJDB7179, DRA accession ID: DRA007456 and GenBank accession numbers: OQ852734-OQ852742. The source data are provided in the source data file. Source data are provided with this paper.

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

## Acknowledgements

We thank Ueli Grossniklaus, Andreas Wagner, Barbara Mable, Vincent Castric, Yuko Wada, and Risa Kobayashi for contributing to the discussion and the Functional Genomics Center Zurich for providing technical support. This study was supported by the Swiss National Science Foundation 310030_212551, 31003A_182318, 31003A_159767, 31003A_140917 to KKS, 310030_212674 to R.S.-I.; the Japan Science and Technology Agency CREST (grant number JPMJCR16O3), Japan; SystemsX.ch SXPHX0-124233; UZH Global Strategy and Partnerships Funding Scheme of the University of Zurich to KKS; the University Research Priority Program of Evolution in Action of University of Zurich to KKS and RSI; the MEXT JSPS KAKENHI (grant numbers 16H06469, 16K21727, 22H02316 to KKS; 16H06467 and 17H05833 to TT; International Leading Research: 22K21352 to TT and KKS; 22H05172 and 22H05179 to KKS and MW; 21H02162 to MW; 21H04711 and 21H05030 to ST); and an EMBO long-term fellowship and a JSPS postdoctoral fellowship for research abroad to TT.

## Author contributions

C-L.Y., T.T., and K.K.S. conceived and designed the study; C.-L.Y., H.K., and T.O. performed the experiments and generated the data supported by R.S.-I., K.S., M.W., and K.K.S.; C-L.Y. analyzed the data with help from T.T., M.H., S.Y., S.T, and K.K.S.; C.-L.Y. and K.K.S. wrote the manuscript. All authors discussed the results and commented on the manuscript.

## Competing interests

The authors declare no competing interests.
