## [Peer Review File · Nature Communications]

REVIEWER COMMENTS

Reviewer #1 (Remarks to the Author):

In this study, the authors studied self-compatible allotetraploid *Arabidopsis kamchatica* accessions to test the evolutionary prediction that dominant mutations are responsible for the transition from self-incompatibility to self-compatibility in this polyploid species (Haldane's sieve). To address this, they took advantage of the complexities of this self-incompatibility system which is controlled by different S-haplogroups that can share dominant-recessive relationships due to epigenetic modifiers. The sRNA modifiers originate from a dominant S-haplogroup to repress the expression of the male SCR gene in the recessive S-haplogroup. Through a series of elegantly designed experiments, Yew et al provide convincing evidence that a mutation in the dominant S-haplogroup is responsible for loss of self-incompatibility in this allotetraploid. Furthermore, they demonstrate that sRNAs from a dominant S-haplogroup in one subgenome are responsible for suppressing the recessive S-haplogroup in the other subgenome showing that this trait has remained intact during polyploidization which would play an important role in enabling the transition to self-compatibility. Overall, the manuscript is well written, and the data presented is very high quality, supporting the manuscript's conclusions. The results are significant as they not only increase our understanding of the role of sRNA modifiers in the *Arabidopsis* self-incompatibility system, but more generally, reveal how interactions between subgenomes can shape the evolution of polyploids.

I do not have any additional experiments that I feel are required, but below are points that do need to be clarified in the text and/or figure legends:

1) Lines 111 -113: "genomic sequence data of *A. kamchatica* bearing haplogroup D to BAC sequences of S-haplogroup Ah12 (=AkD) suggested a deletion in the chromosomal region encompassing AkSCR (Fig. S4)".

- Please describe how much of AkSCR-D is predicted to be deleted. Presumably, this is the explanation for no expression of AkSCR-D in Fig 1c (rather than the impact of mirS2)

- Is the AkSCR-D promoter still present in the genome as the authors describe AkBmirS2 targeting the SCR-D promoter (line 125) (Fig S6 and S7), and where is the SCR-D promoter in Fig S4B? The gene orientations are not shown.

2) Lines 129-131: "The 24-nt sRNA of AkAmirS3 was found in the AD (Takashima) accession, but not that of AkAmirS2, which was not detected despite the presence of its precursor (Fig. 1d, Fig. S6b and 8b)"

- Is AkAmirS3 and AkAmirS2 are reversed in this text or is Fig 1d labelled incorrectly? Fig 1d shows AkAmirS2 expression (top-panel) and no data is shown for AkAmirS3.

- Are these sRNAs also predicted to target the SCR-D promoter (AkAmirS2) and SCR-E intron (AkAmirS3)?

- These are very important points to clarify since the experiment in Figure 4 is based on the premise that AkAmirS2 is not expressed in the Takashima accession (AD heterozygote) and therefore cannot suppress the promoter of the recessive SCR-D transgene.

3) Lines 119-121: "(Fig. S5-8, named AkBmirS2, AkBmirS3, AkAmirS2, AkAmirS3). They were expressed in the anthers and co-segregated with SCR sequences at the S-locus (Fig 1d, Supplementary Table 1)."

- a minor correction to this statement since either AkAmirS3 or AkBmirS2 is not expressed (depending on the correction in point #2) and there no segregation data in Table S1.

4) In Fig 1C and Fig S1a, the *A. halleri* plants are labelled as A, B or D. Are these plants homozygous for the A, B or D haplogroups? Or are they heterozygous with other *A. halleri* S-haplogroups?

Reviewer #2 (Remarks to the Author):

This manuscript by Yew et al. investigates the mechanism of the transition from self-

incompatibility to self-compatibility in *Arabidopsis kamchatica* and one of its diploid relatives *A. halleri*. This manuscript would be of interest to the larger communities interested in both polyploid evolution and the evolution of self-compatibility, and the authors present novel evidence for the mechanisms of the loss of the S-locus, a TE insertion. While I have minimal critiques of the work presented, I have concerns that the framing of the work as a novel idea that this self-compatibility mechanism “supports Haldane’s sieve” is not a novel finding. Additionally, the scholarship lacks context in relation to recent work on the evolution of self-compatibility, particularly in Brassicaceae and in *Arabidopsis kamchatica*, and lacks the proper framework of polyploid formation, epigenetics, and hybridization that would be expected.

The introduction of this manuscript is lacking in context and completeness. The opening line of the introduction suggests that polyploids “may suffer from a ‘retarded’ phenotypic evolution” which lies in stark contrast to the body of work that suggests that polyploids often have faster ecological niche differentiation, are more robust to stress, and are physically larger than their diploid relatives, among other well-described phenotypic differences. The second and third paragraphs contain 2 and 1 sentences, respectively, despite there being obvious avenues for improved scholarship (for instance, “genome shock” is described as a single sentence in the second paragraph, but could easily be expanded to an entire paragraph to better reflect the current knowledge of the field).

Additionally, there is no mention of two important, recent manuscripts on self-compatibility mechanisms in the Brassicaceae, specifically in *Arabidopsis kamchatica*. Firstly, the authors should include in their scholarship Novikova et al., 2022, *Plant Reproduction* (<https://doi.org/10.1007/s00497-022-00451-6>), which reviews the role of self-compatibility in the formation of allopolyploids, and in particular highlights the mechanisms of allopolyploid formation in four Brassicaceae allopolyploids including *Arabidopsis kamchatica*. This work also highlights that sporophytic self-compatibility via SCR/SP11 is rarely codominant, indicating that the dominance of self-compatibility is not a novel finding. As such, the title of this manuscript could be easily, yet incorrectly, interpreted as the first evidence that self-compatibility systems supports the hypothesis of Haldane’s sieve.

Secondly, the authors should consider including the work from Kolesnikova et al., (<https://doi.org/10.1101/2022.06.24.497443>), which also describes the genomic basis of self-compatibility in *Arabidopsis kamchatica* its diploid relatives. Because this work is currently only available as a preprint on bioRxiv, I do not believe that its publication reduces the quality of this manuscript in any way, and does not influence my review or described justifications for any publication decision. However, the inclusion of this work in the current manuscript would add to the scholarship presented here, and would provide a better framework for the reader to understand the results.

Finally, I have provided some additional comments, with line-numbers included.

Line 44: How does the permanent heterozygosity in allopolyploids affect the masking effect via genetic dominance?

Line 49: I think you mean “between homoeologous copies in polyploid species” which would be more precise than “duplicated homologous copies”.

Line 85: Does this imply that D is dominant over E?

Line 201: Are there no studies that suggest that there are other important epigenetic factors besides “genomic shock”? It appears that you are presenting only one side of this topic.

Line 206-207: I don’t understand this clause: “genome-wide changes at polyploidization events are not necessary or are even deleterious”. What does “not necessary” mean in this context?

Line 293/413: In the methods, you describe that all pollen types (A,B,D) were applied to the stigmas of all maternal haplotypes (AB, AD, BD), but no results are shown from the AB haplotype, not for other combinations. Why is this?

Line 323: What were the 'default parameters' for bowtie?

Line 323: There are newer algorithms for sRNA detection (e.g. shortStack) that can do this in a more sophisticated way than visually using IGV. Why were these methods not used?

Reviewer #3 (Remarks to the Author):

Self-incompatibility in Arabidopsis is controlled by S-locus encoding a pair of interacting proteins, SCR/SP11 specifically expressed in males and SRK specifically expressed in females. Self-incompatibility is collapsed in the allotetraploid *A. kamchatica* containing two S-loci derived from self-incompatible parents. Three accessions of *A. kamchatica* are determined according to the S-haplogroups, Takashima with AD, Potter with BD, and Okhotsk with BE. By pair-wise examination, the authors determined that SCR-B is the dominant SCR, and that introduction of functional SCR-B restores self-incompatibility. However, the evidence for the final conclusion "small RNA regulate dominance of self-incompatibility" is not convincing.

The logic behind the author's final conclusion is as follows:

First, introduction of AhSCR-D into Potter accession (BD haplotype) under the control of its native promoter showed no transcription and failed to restore self-incompatibility.

Second, introduction of the same construct to Takashima accession (AD haplotype) restored self-incompatibility.

Third, given the lack of 24-nt mirS2 sRNA in takashima, the author reached the conclusion that small RNA regulate self-incompatibility.

The logical flaw is that there is no strict control to draw conclusions. There may be other genetic differences between two accessions. The different output of self-incompatibility may be due to other genetic differences between accessions.

Further transgenic evidence is needed to draw a solid conclusion.

1. Knock out the precursor of 24-nt mirS2 sRNA in Potter, and see whether the introduction of native promoter-driven AhSCR-D can restore self-incompatibility
2. Introducing 24-nt mirS2 sRNA or precursor in Takashima to see if transgenic AhSCR-D driven by its native promoter no longer restore self-incompatibility.
3. The typical role of 24nt sRNAs is to mediate RNA-dependent DNA methylation, and the authors needed to demonstrate the DNA methylation status in the putative sRNA-targeted promoter region in compatible and incompatible hybrids.

Minor:

170-172. "These findings support the hypothesis that the insertion of the transposable element in SCR-B conferred a dominant self-compatible mutation."

There is a jump here as this section describes experiments involving SCR-D rather than SCR-B.

Dear Reviewers,

Thank you for your kind response.

We appreciated the constructive comments from the reviewers and thoroughly revised the manuscript as suggested. Furthermore, we added new data on dominance (new Supplementary Fig. 1a).

Our point-to-point responses are shown in Arial font starting with >. Typos were corrected by a language service.

RESPONSE TO REVIEWER COMMENTS

Reviewer #1 (Remarks to the Author):

In this study, the authors studied self-compatible allotetraploid *Arabidopsis kamchatica* accessions to test the evolutionary prediction that dominant mutations are responsible for the transition from self-incompatibility to self-compatibility in this polyploid species (Haldane's sieve). To address this, they took advantage of the complexities of this self-incompatibility system which is controlled by different S-haplogroups that can share dominant-recessive relationships due to epigenetic modifiers. The sRNA modifiers originate from a dominant S-haplogroup to repress the expression of the male SCR gene in the recessive S-haplogroup. Through a series of elegantly designed experiments, Yew et al provide convincing evidence that a mutation in the dominant S-haplogroup is responsible for loss of self-incompatibility in this allotetraploid. Furthermore, they demonstrate that sRNAs from a dominant S-haplogroup in one subgenome are responsible for suppressing the recessive S-haplogroup in the other subgenome showing that this trait has remained intact during polyploidization which would play an important role in enabling the transition to self-compatibility. Overall, the manuscript is well written, and the data presented is very high quality, supporting the manuscript's conclusions. The results are significant as they not only increase our understanding of the role of sRNA modifiers in the *Arabidopsis* self-incompatibility system, but more generally, reveal how interactions between subgenomes can shape the evolution of polyploids.

I do not have any additional experiments that I feel are required, but below are points that do need to be clarified in the text and/or figure legends:

1) Lines 111 -113: "genomic sequence data of *A. kamchatica* bearing haplogroup D to BAC sequences of S-haplogroup Ah12 (=AkD) suggested a deletion in the chromosomal region encompassing AkSCR (Fig. S4)".

- Please describe how much of AkSCR-D is predicted to be deleted. Presumably, this is the explanation for no expression of AkSCR-D in Fig 1c (rather than the impact of mirS2)

>Thank you for pointing this out.

The data suggest that *AkSCR-D* is entirely lacking. The SCR sequences were isolated using the diploid *A. halleri* and used in subsequent analyses (for example, Fig. S6). We modified the main text as follows.

Old: Southern blotting analysis and read mapping of high-throughput genomic sequence data of *A. kamchatica* bearing haplogroup D to BAC sequences of S-haplogroup Ah12 (=AkD) suggested a deletion in the chromosomal region encompassing *AkSCR-D* (Fig. S4).

New: Southern blotting analysis and read mapping of high-throughput genomic sequence data of *A. kamchatica* bearing haplogroup D to BAC sequences of S-haplogroup Ah12 (=AkD) suggested deletion in the chromosomal region encompassing *AkSCR-D*

(Supplementary Fig. 4), resulting in the lack of *AkSCR-D* expression. Due to the lack of *SCR-D* sequence in *A. kamchatica*, that of *A. halleri* is used in the following analyses.

- Is the *AkSCR-D* promoter still present in the genome as the authors describe *AkBmirS2* targeting the *SCR-D* promoter (line 125) (Fig S6 and S7), and where is the *SCR-D* promoter in Fig S4B? The gene orientations are not shown.

>We agree with you; our explanation was indeed not adequate. We noticed that we explained it only within Fig. 1e ("*AhSCR-D* promoter"). The *SCR-D* sequence in Fig. S6 is from *A. halleri*, not from *A. kamchatica*. We have now revised Fig. S4b and its legend to show the coding regions and the gene directions, and we hope it is now clear that the deletion includes the promoter regions.

Supplementary Fig. 4b

Old: The locations of the *SCR-D* and *SRK-D* genes on the BAC sequences are indicated by yellow and purple bars, respectively.

New: The locations of the *SCR-D* (<1543...>2413) and *SRK-D* (<7972..11899) genes corresponding to predicted coding sequences on the BAC sequences are indicated by yellow and purple arrows, respectively. The direction of the arrow indicates the gene direction.

In addition, we modified the legend of Fig. S6 to explicitly indicate that the promoter sequence of *A. halleri* was used. We believe Fig. S7 is not directly related to this point but still updated the legend of Fig. 1e.

New: Fig. S6c. Potential target site of *AkBmirS2* at the promoter of the recessive *SCR-D* obtained from *A. halleri*.

2) Lines 129-131: "The 24-nt sRNA of *AkAmirS3* was found in the AD (Takashima) accession, but not that of *AkAmirS2*, which was not detected despite the presence of its precursor (Fig. 1d, Fig. S6b and 8b)"

- Is *AkAmirS3* and *AkAmirS2* are reversed in this text or is Fig 1d labelled incorrectly? Fig 1d shows *AkAmirS2* expression (top-panel) and no data is shown for *AkAmirS3*.

>Thank you for your comments.

The labels are correct. We noticed that the notation of precursor RNA and small RNA varies among previous publications and may be confusing. Therefore, in the revised file, we have added "precursor" or "sRNA" in their first appearance and at other instances where it was ambiguous.

New: We found the precursor genes of *mirS2* and *mirS3* sRNAs from each of the two dominant S-haplogroups B and A (Supplementary Figs. 5–8, named *AkBmirS2*, *AkBmirS3*, *AkAmirS2*, and *AkAmirS3* precursor genes).

In addition, we now refer to each figure separately in the sentence for clarity.

Old: The 24-nt sRNA of *AkAmirS3* was found in the AD (Takashima) accession, but not that of *AkAmirS2*, which was not detected despite the presence of its precursor (Fig. 1d, Fig. S6b and 8b). This may indicate a secondary decay of the precursor processing.

New: The 24-nt sRNA of *AkAmirS3* was found in the AD (Takashima) accession (Supplementary Fig. 8b), but that of *AkAmirS2* (Supplementary Fig. 6b) was not detected despite the expression of its precursor gene *AkAmirS2* (Fig. 1d). This may indicate a secondary decay of the precursor processing of *AkAmirS2*

We also modified Figure 1e to emphasize that AkBmirS2 is RNA (U is used instead of T in DNA).

- Are these sRNAs also predicted to target the SCR-D promoter (AkAmirS2) and SCR-E intron (AkAmirS3)?

- These are very important points to clarify since the experiment in Figure 4 is based on the premise that AkAmirS2 is not expressed in the Takashima accession (AD heterozygote) and therefore cannot suppress the promoter of the recessive SCR-D transgene.

>Thank you for your comments.

In brief, the *AkAmirS3* sRNA is expressed in the AD (Takashima) accession, as shown in Fig. S8b. The expression data of the *AkAmirS3* precursor is not presented because there is no natural individual of AE, in which the *AkAmirS3* sRNA from haplogroup A can potentially bind to *SCR-E*. Thus, it is not relevant to the conclusions. *AkAmirS2* sRNA was not detected in Tak (Fig. S6b) even though its precursor was detected (Fig. 1d). We hope these explanations clarify our point.

3) Lines 119-121: "(Fig. S5-8, named AkBmirS2, AkBmirS3, AkAmirS2, AkAmirS3). They were expressed in the anthers and co-segregated with SCR sequences at the S-locus (Fig 1d, Supplementary Table 1)."

- a minor correction to this statement since either AkAmirS3 or AkBmirS2 is not expressed (depending on the correction in point #2) and there no segregation data in Table S1.

>Thank you very much.

Yes, you are right. The new sentence implies that the co-segregation of three of the four was confirmed. As described just above, *AkAmirS3* is not relevant to the conclusions because there is no natural individual of AE, wherein *AkAmirS3* sRNA from haplogroup A can potentially bind to *SCR-E*.

New: We confirmed that the former three of them co-segregated with *SCR* sequences at the S-locus (Supplementary Table 1).

The expression in stamen is now mentioned in later sentences on sRNA.

New: Mapping sRNA reads to the *AkBmirS2* precursor gene from the BD (Potter) and BE (Okhotsk) accessions revealed the presence of a 24-nt sRNA in anthers (Supplementary Fig. 6a)

4) In Fig 1C and Fig S1a, the *A. halleri* plants are labelled as A, B or D. Are these plants homozygous for the A, B or D haplogroups? Or are they heterozygous with other *A. halleri* S-haplogroups?

>Thank you for pointing this out.

The details of the genotypes are described in Table S7. The reasoning is described in the Methods section (*S*-specificity of *A. halleri* used in this experiment was previously confirmed via interspecific crossings with *A. kamchatica*¹⁸).

The phenotypes are as follows.

A: A, X*

B: B, X*

D: D, C

X* Unknown *S*-haplogroup that was phenotypically recessive (C is the most recessive allele).

To clarify the point, we added the following sentence to the legend of Fig. 1a and S1a, while maintaining the visibility of the figures.

New: See Supplementary Table 7 for the genotypes of *A. halleri* individuals.

Reviewer #2 (Remarks to the Author):

This manuscript by Yew et al. investigates the mechanism of the transition from self-incompatibility to self-compatibility in *Arabidopsis kamchatica* and one of its diploid relatives *A. halleri*. This manuscript would be of interest to the larger communities interested in both polyploid evolution and the evolution of self-compatibility, and the authors present novel evidence for the mechanisms of the loss of the S-locus, a TE insertion. While I have minimal critiques of the work presented, I have concerns that the framing of the work as a novel idea that this self-compatibility mechanism “supports Haldane’s sieve” is not a novel finding. Additionally, the scholarship lacks context in relation to recent work on the evolution of self-compatibility, particularly in Brassicaceae and in *Arabidopsis kamchatica*, and lacks the proper framework of polyploid formation, epigenetics, and hybridization that would be expected.

>Thank you for your understanding of our work.

We would like to emphasize that the strength of this manuscript is the provision of transgenic evidence to support the relevance of dominance in allopolyploid species. We do not insist that it is “a novel idea.” In the previously submitted version, we cited many extant papers reporting the existence of small RNA in various allopolyploid species and a theoretical paper suggesting the importance of dominance in the evolution of self-compatibility, including our previous papers. In the revised version, we have provided more details on the previous research to clarify this point.

We suggest that a major gap in the research was caused by the lack of model polyploid species with transgenic techniques.

The title was changed to avoid a misunderstanding.

New title: Dominance in self-compatibility of allopolyploid *Arabidopsis kamchatica* shown by transgenic restoration of self-incompatibility

The introduction of this manuscript is lacking in context and completeness. The opening line of the introduction suggests that polyploids “may suffer from a ‘retarded’ phenotypic evolution” which lies in stark contrast to the body of work that suggests that polyploids often have faster ecological niche differentiation, are more robust to stress, and are physically larger than their diploid relatives, among other well-described phenotypic differences.

>The introduction of the submitted version was formatted as a short report. We have completely revised it and provided more details.

First, we discussed classic literature (Stebbins 1971), followed by more recent views. We revised the sentences to now clearly indicate that this is a historical view.

The second and third paragraphs contain 2 and 1 sentences, respectively, despite there being obvious avenues for improved scholarship (for instance, “genome shock” is described as a single sentence in the second paragraph, but could easily be expanded to an entire paragraph to better reflect the current knowledge of the field).

>These paragraphs have now been extended.

New: The role of epigenetics in polyploid evolution has long been discussed^{3,4,22}. In the synthetic allopolyploid plants of many species, genome-wide instability in gene expression and methylation has been reported and termed as “genome shock”^{3,4,22,23}. It has been debated whether novel genetic and epigenetic variants contributed to adaptive evolution of polyploids or if were deleterious^{3,24,25}. Besides genome-wide studies on small RNA (sRNA) of allopolyploid species^{26–28}, the dominance relationship of self-incompatibility in Brassicaceae regulated by sRNA has also been studied in polyploid species^{2,11,19}.

Additionally, there is no mention of two important, recent manuscripts on self-compatibility mechanisms in the Brassicaceae, specifically in *Arabidopsis kamchatica*. Firstly, the authors should include in their scholarship Novikova et al., 2022, Plant Reproduction (<https://doi.org/10.1007/s00497-022-00451-6>), which reviews the role of self-compatibility in the formation of allopolyploids, and in particular highlights the mechanisms of allopolyploid formation in four Brassicaceae allopolyploids including *Arabidopsis kamchatica*. This work also highlights that sporophytic self-compatibility via SCR/SP11 is rarely codominant, indicating that the dominance of self-compatibility is not a novel finding. As such, the title of this manuscript could be easily, yet incorrectly, interpreted as the first evidence that self-compatibility systems supports the hypothesis of Haldane’s sieve.

>Thank you for this suggestion. We have cited this paper in our revised manuscript. In the previously submitted version, we had already cited many papers and reviews describing the idea of dominance in self-compatibility and Haldane's sieve (including our previous publications such as Tsuchimatsu et al. 2012, Shimizu and Tsuchimatsu 2015), and therefore we did not intend to claim this to be a novel idea. As described above, we changed the title to avoid a misunderstanding.

New title: Dominance in self-compatibility of allopolyploid *Arabidopsis kamchatica* shown by transgenic restoration of self-incompatibility

Secondly, the authors should consider including the work from Kolesnikova et al., (<https://doi.org/10.1101/2022.06.24.497443>), which also describes the genomic basis of self-compatibility in *Arabidopsis kamchatica* its diploid relatives. Because this work is currently only available as a preprint on bioRxiv, I do not believe that its publication reduces the quality of this manuscript in any way, and does not influence my review or described justifications for any publication decision. However, the inclusion of this work in the current manuscript would add to the scholarship presented here, and would provide a better framework for the reader to understand the results.

>We cited the Preprint at three points in the introduction and the discussion. The Preprint is focused on sequencing and is mostly complementary to our manuscript focusing on functional experiments. We mentioned it in the context of the importance of experimental evidence as follows.

Citation number 53

Introduction:

Both progenitor species are predominantly self-incompatible, although self-compatible individuals of *A. lyrata* are reported in North America⁵² and Siberia^{50,53,54}

Discussion:

Alternatively, the mutations may have been segregated or fixed in diploid progenitor species, considering that the transition to self-compatibility is among the most frequent evolutionary transitions in angiosperms^{7,11,50,53}

Discussion:

Co-dominance of S-haplogroups B and D was observed in pistils¹⁸, and a previous report pointed out that S-haplogroup D belongs to the most dominant "phylogenetic class"⁵³ based on phylogenetic analysis of *SRK* sequences³³. However, our crossing experiment showed the male dominance of B over D.

Finally, I have provided some additional comments, with line-numbers included.

Line 44: How does the permanent heterozygosity in allopolyploids affect the masking effect via genetic dominance?

>The original sentence was too short and ambiguous. We revised the entire introduction, including this point.

"Theoretical studies have suggested that additional alleles or duplicated copies may confer a masking effect on recessive and additive mutations because of redundancy, and therefore a mutation in each duplicated gene may need to be accumulated in allopolyploid species for phenotypic evolution such as the evolution of self-compatibility^{3,4,16,17}. Dominance relationships can reconcile these two aspects of polyploidy by alleviating the masking effect^{2,11,18,19}. Haldane suggested that dominant or partially dominant mutations contribute more to adaptive evolution than recessive mutations, which is referred to as Haldane's sieve²⁰. The principle can be valid for both allelic interactions in diploid species and epistatic interactions between homologous copies in polyploid species because allotetraploid species are effectively fixed heterozygotes of subgenomes derived from different species."

Line 49: I think you mean "between homoeologous copies in polyploid species" which would be more precise than "duplicated homologous copies".

>Thank you. We have changed this as suggested.

Line 85: Does this imply that D is dominant over E?

>Thank you for the comment. We have now explained the notation "the S-haplogroup B was dominant over D (noted as B > D)" before stating "D (= Ah12) > E (= Ah02)"

Line 201: Are there no studies that suggest that there are other important epigenetic factors besides "genomic shock"? It appears that you are presenting only one side of this topic.

>In the introduction of the revised version, a new paragraph is added explaining the epigenetics of polyploid species including small RNA and massive stochastic changes with genome shock. The revised discussion does not mention genome shock. The revisions made in the introduction are provided below.

New: The role of epigenetics in polyploid evolution has long been discussed^{3,4,22}. In the synthetic allopolyploid plants of many species, genome-wide instability in gene expression and methylation has been reported and termed as "genome shock"^{3,4,22,23}. Further, it has been debated if novel genetic and epigenetic variants contributed to adaptive evolution of polyploids or if were deleterious^{3,24,25}. Besides genome-wide studies of small RNA (sRNA) of allopolyploid species²⁶⁻²⁸, the dominance relationship of self-incompatibility in Brassicaceae regulated by sRNA was also studied in polyploid species^{2,11,19}.

Line 206-207: I don't understand this clause: "genome-wide changes at polyploidization events are not necessary or are even deleterious". What does "not necessary" mean in this context?

>This sentence was removed from the revised text.

Line 293/413: In the methods, you describe that all pollen types (A,B,D) were applied to the stigmas of all maternal haplotypes (AB, AD, BD), but no results are shown from the AB haplotype, not for other combinations. Why is this?

>Thank you for your comment.

We have now added additional results in Supplementary Fig. 1a to show the incompatibility reaction of all combinations using the AB genotype, and we modified the Results and Methods. In the original submission, the individual AB was used only as a control in expression analysis (current Supplementary Fig. 1b). In the revised version, we added further expression data of *A. kamchatica* (current Supplementary Fig. 1c). Although the new data serve as a control experiment and do not affect the main conclusions, we hope that they strengthen the manuscript.

New sentence in Results: In addition, A and B were co-dominant (A = B) (Supplementary Fig. 1a).

New Supplementary Fig. 1a.

Supplementary Fig. 1. Dominance and expression of SCR genes in the anther cDNAs of *Arabidopsis halleri* and *A. kamchatica* bearing different S-haplogroups and alignments of amino acid sequences. a. Crossings were performed to investigate the dominance hierarchy of S-haplogroups A, B, and D. Numerators denote crosses in which more than 20 pollen tubes penetrated the stigma, indicating a compatible reaction. Denominators denote the total number of crosses conducted in each combination. NP: not performed. See Supplementary Table 7 for the genotypes of *A. halleri* individuals.

Methods, Section **Dominance hierarchy of S-haplogroups A, B, and D**

To test the dominance relationship between A and D in pollen, the pollen grains of *A. halleri* individuals with AD were applied to the stigmas of *A. halleri* bearing haplogroup A and that bearing haplogroup D. Similarly, to test the dominance relationship between B and D in pollen, the pollen grains of *A. halleri* individuals with BD were crossed to stigmas bearing haplogroup B and those bearing haplogroup D. To test the dominance relationship between A and B, in addition to stigmas, the pollen grains of *A. halleri* individuals with AB were crossed to the stigmas bearing haplogroup A, B, as well as D, the last of which was performed to show that the pollen grains were viable in this co-dominant case. Pollen tube growth was examined as described¹⁸. S-specificity of *A. halleri* used in this experiment was previously confirmed via interspecific crossings with *A. kamchatica*¹⁸. The expression of *SCR-A*, *SCR-B*, and *SCR-D* were also examined using these genotypes.

Line 323: What were the 'default parameters' for bowtie?

>We added the command to the first appearance, which is fairly simple, and then we referred to it later.

command: bowtie2 --no-unal -x [file name of reference fasta] -U [file name of fastq]⁶⁴

using Bowtie 2.3.2⁶⁴ and viewed in the IGV 2.3.35⁶⁵ as described above

BWA 0.7.12 (command: bwa mem -t8 [file name of reference fasta] [file name of fastq])⁶⁷ and IGV 2.3.35⁶⁵ were similarly used for the mapping and the visualization of genomic sequences (SAMD00045705, SAMD00089932, SAMD00089933)^{50,68} on the BAC sequences of the S-locus region of S-haplogroup Ah12 (Genbank KJ772374.1)³³.

Line 323: There are newer algorithms for sRNA detection (e.g. shortStack) that can do this in a more sophisticated way than visually using IGV. Why were these methods not used?

>Our methods clearly support the existence of target sRNAs. We also suggest it is sufficient for the conclusion.

Reviewer #3 (Remarks to the Author):

Self-incompatibility in *Arabidopsis* is controlled by S-locus encoding a pair of interacting proteins, SCR/SP11 specifically expressed in males and SRK specifically expressed in females. Self-incompatibility is collapsed in the allotetraploid *A. kamchatica* containing two S-loci derived from self-incompatible parents. Three accessions of *A. kamchatica* are determined according to the S-haplogroups, Takashima with AD, Potter with BD, and Okhotsk with BE. By pair-wise examination, the authors determined that SCR-B is the dominant SCR, and that introduction of functional SCR-B restores self-incompatibility. However, the evidence for the final conclusion "small RNA regulate dominance of self-incompatibility" is not convincing.

>We thank you for the summary; however, we did not intend to make the conclusion, "small RNA regulate dominance of self-incompatibility." We realize that the title "Small RNA-regulated dominance among polyploid subgenomes supports Haldane's sieve in the evolution of self-compatibility" may have been misleading. This short title, "small RNA-regulated dominance," was intended to give the background information of previous studies.

We now revised the title to avoid misunderstanding.

New title: Dominance in self-compatibility of the allopolyploid *Arabidopsis kamchatica* shown by transgenic restoration of self-incompatibility

This manuscript focused on providing transgenic evidence of dominance in polyploid species. We propose that transgenic experiments of small RNA can be an important topic for future research; however, the functional validation of sRNA in dominance has already been rigorously tested in diploid species and is beyond the scope of this manuscript.

We believe that the focus on dominance but not sRNA was already specified throughout the manuscript. To make this point explicitly clear, we have now added the following sentences to the discussion.

"In this study, we focused on the transgenic experiments of SCR but did not manipulate sRNA. The epigenetic regulation of SCR expression by sRNA is supported by transgenic

experiments in diploid self-incompatible species, and therefore we suggest the importance of epigenetic regulation in the dominance of the allopolyploid *A. kamchatica*."

Furthermore, we added a paragraph in the Introduction and two in the Discussion to explain the importance of transgenic evidence in studying genes responsible for self-compatibility, and we emphasized the motivations of this study in the last paragraph of the Introduction.

New in the Introduction: "Functional experiments are important to identify the genes responsible for the evolution of self-compatibility because sequence analysis alone may be confounded by mutations that are difficult to be detected such as expression loss and by secondary degrading mutations in other genes in the self-incompatibility system after the loss of self-incompatibility^{11,56}. In *A. thaliana*, the standard accession Col-0 had a gene disruptive mutation both in *SCR* and *SRK*⁵⁷ but several accessions including Wei-1 have a full-length *SRK*⁵⁸. By the transgenic introduction of a repaired *SCR*, *A. thaliana* Wei-1 restored self-incompatibility⁵⁵. Such transgenic restoration by introducing a single gene shows that the mutation in the gene is responsible for self-compatibility and that all other genes for self-incompatibility are functional. The self-incompatibility system of Brassicaceae involves many genes other than *SCR* and *SRK*, and therefore self-compatibility may be attributed to so-called modifier genes such as *MLPK* that are not linked to *S*-locus³⁶, or to an unknown locus in the North American *A. lyrata*⁵². Because the sRNAs regulating male dominance are located at the *S*-locus^{29,31}, the self-compatibility mutation must be located at the *S*-locus for dominance to be linked.

In this study, we experimentally examined the male dominance relationships using the self-incompatible diploid species. Based on the sequence analysis of *SCR* and sRNA sequences in *A. kamchatica*, we experimentally restored the *SCR* sequences of *S*-haplogroups B and D by transgenic experiments. Using them, we tested whether the male self-incompatibility specificity gene *SCR* was responsible for the evolution of self-compatibility and whether restoration of the dominant but not the recessive *S*-haplogroup can confer self-incompatibility."

New in the Discussion: "In this study, we first found the dominance of haplogroups A/B over D in pollen, and identified *SCR* sequences and sRNA. Transgenic plants with repaired *SCR* of the dominant *S*-haplogroup B restored the self-incompatibility. Similar to a previous transgenic study of the diploid natural species *A. thaliana*⁵⁵, our results indicate that the mutation in the male specificity gene was responsible for self-compatibility also in *A. kamchatica*, supporting the adaptive spread of male self-compatible mutations predicted by the theory of sexual asymmetry¹². Furthermore, our transgenic experiment showed that the self-compatible mutation was located at the *S*-locus, which encompasses the sRNAs responsible for male dominance. By contrast, the self-incompatibility was not restored when the recessive *S*-haplogroup D was repaired in the same accession, although this construct turned out functional in another accession that lacked the *mirS2* sRNA. Together, the data suggest that a single loss-of-function mutation in the self-incompatibility specificity gene *SCR-B* conferred dominant self-compatibility (Supplementary Fig. 11)."

"Our study highlighted the importance of experimental evidence in addition to sequencing for clarifying complex dominance mechanisms in the Brassicaceae self-incompatibility. The co-dominance of *S*-haplogroups B and D was observed in pistils¹⁸, and a previous report pointed out that *S*-haplogroup D belongs to the most dominant "phylogenetic class"⁵³ based on phylogenetic analysis of *SRK* sequences³³. However, our crossing experiment showed the male dominance of B over D. Furthermore, the evolutionary reversal experiment of *SCR-B* showed that the mutation in *SCR* was responsible for the evolution of self-compatibility, not that in other known or unknown genes. In this study, we focused on the transgenic experiments of *SCR* but did not manipulate sRNA. The epigenetic regulation of *SCR* expression by sRNA was also supported by transgenic experiments in diploid self-

incompatible species²⁹, and therefore we also suggest the importance of epigenetic regulation in the dominance of the allopolyploid *A. kamchatica*.

The *S*-locus inherited from *A. halleri* and another inherited from *A. lyrata* were combined in *A. kamchatica*¹⁸. Our study suggested that the *S*-haplogroup located on one of the two subgenomes was dominant over another *S*-haplogroup located on another subgenome. Analogous to the epigenetic regulation by sRNAs in diploid heterozygotes, our data lead to a hypothesis of epigenetic regulation between subgenomes."

The logic behind the author's final conclusion is as follows:

First, introduction of *AhSCR-D* into Potter accession (BD haplotype) under the control of its native promoter showed no transcription and failed to restore self-incompatibility.

Second, introduction of the same construct to Takashima accession (AD haplotype) restored self-incompatibility.

Third, given the lack of 24-nt *mirS2* sRNA in takashima, the author reached the conclusion that small RNA regulate self-incompatibility.

The logical flaw is that there is no strict control to draw conclusions. There may be other genetic differences between two accessions. The different output of self-incompatibility may be due to other genetic differences between accessions.

>Thank you for your comments.

We agree that there may be other genetic differences, which is why we clearly explained the motivation for this experiment (Line 158-160 of the first submission in the section No restoration of self-incompatibility by repairing a mutation in the recessive *S*-haplogroup) as "however, it does not rule out the possibility that the *AhSCR-D* genomic fragment is nonfunctional owing to the lack of some regulatory elements. To confirm that the *AhSCR-D* construct can confer self-incompatibility..." To clarify this point, we have revised the sentences as follows:

"To confirm that the *AhSCR-D* construct is adequate to confer self-incompatibility, it was introduced into the Takashima accession. Consistent with the lack of the 24-nt *mirS2* sRNA in this accession, the expression of *SCR-D* was detected in all seven transgenic lines."

"This experiment showed that the *AhSCR-D* genomic fragment is adequate to confer self-incompatibility in this accession."

Further transgenic evidence is needed to draw a solid conclusion.

1. Knock out the precursor of 24-nt *mirS2* sRNA in Potter, and see whether the introduction of native promoter-driven *AhSCR-D* can restore self-incompatibility
2. Introducing 24-nt *mirS2* sRNA or precursor in Takashima to see if transgenic *AhSCR-D* driven by its native promoter no longer restore self-incompatibility.
3. The typical role of 24nt sRNAs is to mediate RNA-dependent DNA methylation, and the authors needed to demonstrate the DNA methylation status in the putative sRNA-targeted promoter region in compatible and incompatible hybrids.

>These experiments support the relevance of the small RNA; however, this is beyond the scope of this study as explained previously. We considered the third experiment; however, technically, the examination of DNA methylation status is possible only using *Brassica*, which has a large stamen, but is much harder when using small *Arabidopsis*.

We would like to point out that conducting the transgenic experiment in a new emerging model species was a major challenge. At least two years is needed to obtain the results of a transgenic line of *A. kamchatica* because of the lower efficiency in the transformation and longer generation time compared with the model species *A. thaliana*.

We suggest that the transgenic experiments of allopolyploid species provide new, strong evidence of the importance of dominance in mating systems. The data not only support previous hypotheses suggested by theoretical studies and sequencing studies but also open the way for functional studies of the interaction of subgenomes in allopolyploid species.

Minor:

170-172. “These findings support the hypothesis that the insertion of the transposable element in SCR-B conferred a dominant self-compatible mutation.”

There is a jump here as this section describes experiments involving SCR-D rather than SCR-B.

>We agree that this sentence is not suitable here and has been deleted.

REVIEWER COMMENTS

Reviewer #1 (Remarks to the Author):

The Yew et al manuscript is a revised version of their study using transgenics to investigate the prediction that the transition from self-incompatibility to self-compatibility in the polyploid *Arabidopsis kamchatica* is due to mutations in the dominant S-haplotypes. I have gone through the revised manuscript and the responses to the previous reviews, and I am very satisfied with the changes made in this revised manuscript. The experiments are very elegantly designed to explore the contributions of the primary SI determinants, SCR and SRK, as well as the sRNAs that establish a dominance hierarchy by repressing the expression of SCR genes in the recessive S-haplotypes. The updated introduction and discussion better reflect the background literature and research, and the results section is much more clearly described and explained.

I just have one minor comment to add:

Page 7, line 248: "Except for transgenic line Tak_SCRD_9, which showed the weakest SCR-D expression, all lines recovered self-incompatibility and produced significantly shorter siliques and fewer seeds compared to those in the wild type"

I completely agree that the expression of SCR-D in the Takashima accession does restore self-incompatibility. However, I feel that the conclusion from Figure 4 is more nuanced than this sentence. The data presented shows all lines displaying some degree of a SI phenotype with Tak_SCRD_9 and Tak_SCRD_11 being on the weaker end of the spectrum. For Tak_SCRD_9, 2/6 pistils displayed <20 pollen tubes (Table S6) and the number of seeds per silique were significantly lower compared to WT-Tak (Table S10).

Reviewer #2 (Remarks to the Author):

In this revision, Yew et al. have made considerable progress in responding to the first round of review. Overall I am pleased with the changes that have been made, and appreciate the authors' willingness to clarify areas of the manuscript which were found to be confusing or misleading. Much of the authors' revisions (including an update to the title) appear to distance themselves from the claims that they are explicitly testing the role of sRNAs in self-compatibility, and the roles of allelic dominance as a characteristic unique to self-incompatibility in *A. kamchatica*. While these changes are appreciated and more accurately reflect the work presented in the current manuscript, I find myself conflicted, as these were the primary foci that supported this manuscript's publication in a widely-read journal as Nature Communications. As such, I wonder if these results would be better suited for a more targeted audience, such as publication in a society journal. Additionally, the authors highlight that doing transgenic experiments is difficult (particularly in polyploids), but this paper is neither the first to perform transgenics on a polyploid system (this method was first developed by the authors in 2017, but also see Shan et al, 2018 Mol Ecol Res. <https://doi.org/10.1111/1755-0998.12935> and Przetakiewicz et al, 2004 Cell Mol Biol Lett, for 2 examples) nor the first to use transgenic experiments to explore this particular self-incompatibility system in the genus (see Nasrallah et al., 2002, Science for an example). While this manuscript is the first to use transgenics to explore self-incompatibility in a polyploid, I am left with the question of whether every manuscript that describes a new exploration of transgenic experiments in a polyploid warrants publication in a journal such as Nature Communications. The answer is obviously "no" – however, the decision of where to draw this distinction (and whether this manuscript remains within that category) is unclear to me.

Secondly, the authors argue that the concept of allelic dominance in an allopolyploid is the primary conceptual advancement for this manuscript, but it is unclear what specific new conceptual advancements are provided. Instead, I see this manuscript as simply an additional example of dominance relationships in self-incompatibility systems, and specifically self-incompatibility systems a polyploid (a list that the authors even enumerate in their introduction on lines 73-74, 95, and 99). While the broader implications of dominance and recessivity in polyploids is a largely unexplored area of research, these relationships in self-compatibility systems is one of the best-studied areas where these relationships have been determined. And, while the addition of transgenic experiments is a useful confirmation of these dominance relationships (and, as the

authors suggest, is useful for identifying whether the rest of the self-incompatibility pathway may have secondary mutations that reinforce the phenotype), these dominance relationships can (and have) already been inferred through classical genetic crosses where the phenotype can be clearly measured in the offspring. Thus, I am unable to see the connection between how transgenic experiments can uniquely lead to insights around allelic and epistatic dominance that classical genetic crosses cannot. If the authors wish to claim that transgenic experiments are necessary for the inference of dominant relationships (and that classical genetic crosses are insufficient), they should do so explicitly in their manuscript.

Finally, I've provided a few comments on specific lines for the authors to consider:

Line 64: Once again, I think you mean "homoeologous copies in a polyploid" instead of "homologous copies in a polyploid". Are you intending to include autopolyploids in this description as well? Also, you might consider using the similar language of Conover and Wendel 2022 (MBE; doi: <https://doi.org/10.1093/molbev/msac024>, who describe this concept as "homoeologous epistatic dominance" to differentiate it from allelic dominance.

Line 98: "[...] that has a homology to the A subgenome [...]" I'm not sure what this is trying to say.

Line 134: Is loss of gene expression really a 'mutation'?

Line 273: This also explains the prevalence in diploids, right? Why is this restricted to only polyploids?

Line 288: What is "transient" about this dominance relationship? Are you suggesting that it was initially dominant, but is now not dominant?

Line 290: Is it clear whether the mutation under study was inherited from the diploid, or is it a novel mutation in the allopolyploids lineage? The authors state that there are self-compatible populations of *A. lyrata* in N. America and Siberia (Line 108), but this point isn't clarified.

Reviewer #3 (Remarks to the Author):

The authors addressed most of my concerns. I have no further comments.

Our point-to-point responses are shown in Arial font starting with >. We also corrected a few typos.

RESPONSE TO REVIEWERS' COMMENTS

Reviewer #1 (Remarks to the Author):

The Yew et al manuscript is a revised version of their study using transgenics to investigate the prediction that the transition from self-incompatibility to self-compatibility in the polyploid *Arabidopsis kamchatica* is due to mutations in the dominant S-haplotypes. I have gone through the revised manuscript and the responses to the previous reviews, and I am very satisfied with the changes made in this revised manuscript. The experiments are very elegantly designed to explore the contributions of the primary SI determinants, SCR and SRK, as well as the sRNAs that establish a dominance hierarchy by repressing the expression of SCR genes in the recessive S-haplotypes. The updated introduction and discussion better reflect the background literature and research, and the results section is much more clearly described and explained.

>We are very glad to see the fascinating summary by Reviewer 1.

I just have one minor comment to add:

Page 7, line 248: “Except for transgenic line Tak_SCRD_9, which showed the weakest SCR-D expression, all lines recovered self-incompatibility and produced significantly shorter siliques and fewer seeds compared to those in the wild type”

I completely agree that the expression of SCR-D in the Takashima accession does restore self-incompatibility. However, I feel that the conclusion from Figure 4 is more nuanced than this sentence. The data presented shows all lines displaying some degree of a SI phenotype with Tak_SCRD_9 and Tak_SCRD_11 being on the weaker end of the spectrum. For Tak_SCRD_9, 2/6 pistils displayed <20 pollen tubes (Table S6) and the number of seeds per silique were significantly lower compared to WT-Tak (Table S10).

Thank you for your careful examination. In the last revision, figures and tables were reformatted so that the p-values were now separated in a supporting table, so that the main text should have been updated. To be very explicit, all statistical tests except for the silique length of Tak_SCRD_9 were significant ($P < 0.05$).

Old: Consistent with the lack of the 24-nt *mirS2* sRNA in this accession, the expression of *SCR-D* was detected in all seven transgenic lines. Except for transgenic line Tak_SCRD_9, which showed the weakest SCR-D expression, all lines recovered self-incompatibility and produced significantly shorter siliques and fewer seeds compared to those in the wild type. Tak_SCRD_8 and Tak_SCRD_13 produced nearly no seeds (Fig. 4a–d and Supplementary Table 6).

New: Consistent with the lack of the 24-nt *mirS2* sRNA in this accession, the expression of *SCR-D* was detected in all seven transgenic lines with a variable expression level. Self-incompatibility reaction was fully or partially observed in all the lines (Supplementary Table 6). Significantly shorter siliques and fewer seeds compared to those in the wild type were observed except for the silique length of the transgenic line Tak_SCRD_9, which showed the weakest SCR-D expression (Fig. 4a-d and Supplementary Table 10). Lines with a higher expression level such as Tak_SCRD_8 and Tak_SCRD_13 produced nearly no seeds (Fig. 4a–d and Supplementary Table 6).

Reviewer #2 (Remarks to the Author):

In this revision, Yew et al. have made considerable progress in responding to the first round of review. Overall I am pleased with the changes that have been made, and appreciate the authors' willingness to clarify areas of the manuscript which were found to be confusing or misleading. Much of the authors' revisions (including an update to the title) appear to distance themselves from the claims that they are explicitly testing the role of sRNAs in self-compatibility, and the roles of allelic dominance as a characteristic unique to self-incompatibility in *A. kamchatica*. While these changes are appreciated and more accurately reflect the work presented in the current manuscript, I find myself conflicted, as these were the primary foci that supported this manuscript's publication in a widely-read journal as Nature Communications. As such, I wonder if these results would be better suited for a more targeted audience, such as publication in a society journal. Additionally, the authors highlight that doing transgenic experiments is difficult (particularly in polyploids), but this paper is neither the first to perform transgenics on a polyploid system (this method was first developed by the authors in 2017, but also see Shan et al, 2018 Mol Ecol Res. <https://doi.org/10.1111/1755-0998.12935> and Przetakiewicz et al, 2004 Cell Mol Biol Lett, for 2 examples) nor the first to use transgenic experiments to explore this particular self-incompatibility system in the genus (see Nasrallah et al., 2002, Science for an example). While this manuscript is the first to use transgenics to explore self-incompatibility in a polyploid, I am left with the question of whether every manuscript that describes a new exploration of transgenic experiments in a polyploid warrants publication in a journal such as Nature Communications. The answer is obviously "no" – however, the decision of where to draw this distinction (and whether this manuscript remains within that category) is unclear to me.

> We would like to emphasize that a major goal of molecular biology is to establish which gene is responsible for a particular phenotype (here, self-compatibility) using DNA manipulative experiments. The golden standard to obtain causal evidence is transgenic experiments or genome editing complemented by fine mapping. Correlative evidence such as genome sequence analysis or gene expression studies can raise interesting hypotheses to be tested by DNA manipulation. Our manuscript is interdisciplinary, in that a classic question in evolutionary biology (polyploid self-compatibility) is rigorously tested by molecular genetics. We further emphasized this point in the Introduction.

Old: Functional experiments are important to identify the genes responsible for the evolution of self-compatibility because sequence analysis alone may be confounded by mutations that are difficult to be detected...

New: Functional experiments such as transgenic technique and genome editing are important to identify the genes responsible for the evolution of self-compatibility because sequence analysis alone may be confounded by mutations that are difficult to be detected...

In June 2023, the importance of the genetic loci responsible for self-compatibility was highlighted by a new paper in Nature Communications (Li, Y., Mamonova, E., Köhler, N., van Kleunen, M. & Stift, M. Breakdown of self-incompatibility due to genetic interaction between a specific S-allele and an unlinked modifier. Nat. Commun. 14, 3420, 2023). They reported a non-S-locus mutation in a self-compatible population of a diploid *Arabidopsis lyrata*. So far in any allopolyploid species, no study provided strong evidence to distinguish non-S-locus self-compatible mutations from those at the S-locus, which also harbor small RNAs regulating the dominance. Our study showed that the self-compatible mutation in the allotetraploid *A. kamchatica* is located at the S-locus. Because the dominance regulation by small RNAs is also located at the S-locus, this is a prerequisite for dominance regulation to be linked to self-compatibility. Thus our data support the importance of the dominance relationship in the allopolyploid species. We cited the paper both in Introduction and Discussion and added the following.

Old: North American *A. lyrata*⁵²

New: self-compatible populations of *A. lyrata*^{50,56}

Old: Furthermore, the evolutionary reversal experiment of SCR-B showed that the mutation in SCR was responsible for the evolution of self-compatibility, not that in other known or unknown genes.

New: Furthermore, the evolutionary reversal experiment of SCR-B showed that the mutation in SCR was responsible for the evolution of self-compatibility, not that in other known or unknown genes such as an S-unlinked modifier gene recently reported from a self-compatible population of *A. lyrata*⁵⁶.

The current manuscript did not claim that the transgenic experiments is difficult, or that we were the first, as far as we aware. In a point-to-point response to another reviewer, it was discussed that transgenic experiments in an emerging model species are not adequately reported partly because they can take a few years, but this is not mentioned in the main text. The two references are indeed representative of this topic in the limitation of citation number and thus we included them.

New: *Arabidopsis kamchatica* and other polyploid species with transgenic technique^{57,58} will be valuable to assess the role of stochastic and inherited epigenetic regulations in other adaptive traits.

Secondly, the authors argue that the concept of allelic dominance in an allopolyploid is the primary conceptual advancement for this manuscript, but it is unclear what specific new conceptual advancements are provided. Instead, I see this manuscript as simply an additional example of dominance relationships in self-incompatibility systems, and specifically self-incompatibility systems a polyploid (a list that the authors even enumerate in their introduction on lines 73-74, 95, and 99). While the broader implications of dominance and recessivity in polyploids is a largely unexplored area of research, these relationships in self-compatibility systems is one of the best-studied areas where these relationships have been determined. And, while the addition of transgenic experiments is a useful confirmation of these dominance relationships (and, as the authors suggest, is useful for identifying whether the rest of the self-incompatibility pathway may have secondary mutations that reinforce the phenotype), these dominance relationships can (and have) already been inferred through classical genetic crosses where the phenotype can be clearly measured in the offspring. Thus, I am unable to see the connection between how transgenic experiments can uniquely lead to insights around allelic and epistatic dominance that classical genetic crosses cannot. If the authors wish to claim that transgenic experiments are necessary for the inference of dominant relationships (and that classical genetic crosses are insufficient), they should do so explicitly in their manuscript.

Seeing the comment "while the addition of transgenic experiments is a useful confirmation of these dominance relationships", we suggest that the transgenic evidence is not just a confirmation but that is the central topic of biology in order to prove that a particular gene is responsible for a focal phenotype.

In addition, we suggest that previous "classical genetic crosses" mentioned by the Reviewer can mean diverse types of studies and are not very relevant to our study. As far as we know, previous studies using polyploid species focused primarily on F1 plants to observe self-compatibility reactions and dominance. Fine-mapping using F2 or subsequent generations is necessary to examine which gene is responsible for self-compatibility, but the S-locus is notoriously difficult for fine mapping because tight linkage between *SRK*, *SCR* and small RNAs could not be broken due to recombination suppression. We are aware of no fine mapping studies at the S-locus of polyploid species. Thus here we employed a transgenic approach.

As described above, we clarified this point furthermore as follows.

Old: Functional experiments are important to identify the genes responsible for the evolution of self-compatibility because sequence analysis alone may be confounded by mutations that are difficult to be detected...

New: Functional experiments such as transgenic technique and genome editing are important to identify the genes responsible for the evolution of self-compatibility because sequence analysis alone may be confounded by mutations that are difficult to be detected...

Regarding the point "Secondly, the authors argue that the concept of allelic dominance in an allopolyploid is the primary conceptual advancement for this manuscript, but it is unclear what specific new conceptual advancements are provided.", we did not use this terminology (conceptual advancement) which split concepts and evidence as far as we aware.

As a major advancement of this manuscript, we designed experiments to obtain contrasting results for dominant vs. recessive S-haplogroups, i.e., restoration of self-incompatibility only when the dominant one was reconstructed by transgenic experiments. These DNA-manipulative experiments using the model polyploid *A. kamchatica* provided strong support for a long-standing hypothesis of the relevance of dominance in polyploid self-compatibility. We are glad to see the summary by Reviewer 1;

"The experiments are very elegantly designed to explore the contributions of the primary SI determinants, SCR and SRK, as well as the sRNAs that establish a dominance hierarchy by repressing the expression of SCR genes in the recessive S-haplotypes."

Furthermore, this has a much broader implication to contribute to a conceptual paradigm shift in epigenetics in polyploids. Previously, novel stochastic epigenetic mutations termed "genome shock" was a major focus of polyploid research. Our study provided strong support for inter-subgenome epigenetic interaction at a specific locus (S-locus). This opens a new research direction to study epigenetic regulation that was inherited and merged from progenitor species.

In addition to current texts, we further emphasized this point in the Abstract, Introduction, and Discussion.

Title

Old: Dominance in self-compatibility of allopolyploid *Arabidopsis kamchatica* shown by transgenic restoration of self-incompatibility

New: Dominance in self-compatibility between subgenomes of allopolyploid *Arabidopsis kamchatica* shown by transgenic restoration of self-incompatibility

Abstract: The following sentence was moved to the end of the Abstract for emphasis, and modified as follows.

Old: The dominance regulation between subgenomes inherited from progenitors contrasts with novel epigenetic mutations at polyploidization termed genome shock.

New: The dominance regulation between subgenomes inherited from progenitors contrasts with previous studies on novel epigenetic mutations at polyploidization termed genome shock.

Last sentence of the discussion

New: *Arabidopsis kamchatica* and other polyploid species with transgenic technique^{57,58} will be valuable to assess the role of stochastic and inherited epigenetic regulations in other adaptive traits.

Finally, I've provided a few comments on specific lines for the authors to consider:

Line 64: Once again, I think you mean "homoeologous copies in a polyploid" instead of "homologous copies in a polyploid". Are you intending to include autopolyploids in this description as well? Also,

you might consider using the similar language of Conover and Wendel 2022 (MBE; doi: <https://doi.org/10.1093/molbev/msac024>, who describe this concept as “homoeologous epistatic dominance” to differentiate it from allelic dominance.

>Thank you very much. We corrected the typo.

Line 98: “[...] that has a homology to the A subgenome [...]” I’m not sure what this is trying to say.

Thank you for the comment. We clarified the sentence.

Old: In *Capsella bursa-pastoris*, the B subgenome has loss-of-function mutations in *SCR* as well as mirS3 that has a homology to the A subgenome, although its *SCR* sequence has not been found⁴².

New: In *Capsella bursa-pastoris*, the B subgenome has loss-of-function mutations in *SCR* as well as mirS3 that has a potential binding site in the A subgenome, although its relevance in *SCR* dominance regulation is unclear due to the lack of full-length *SCR* sequence in the A genome⁴².

Line 134: Is loss of gene expression really a ‘mutation’?

> The sentence was modified to explicitly mean mutations in regulatory regions.

Old: because sequence analysis alone may be confounded by mutations that are difficult to be detected such as expression loss

New: because sequence analysis alone may be confounded by mutations that are difficult to be detected such as those in regulatory regions resulting in expression loss

Line 273: This also explains the prevalence in diploids, right? Why is this restricted to only polyploids?

In this paragraph, polyploids were first discussed and then followed by diploids, so it would be clear we do not restrict to polyploids. However, to further clarify, we modified the sentence as follows.

Old: The high frequency of loss-of-function of *SCR* or *SRK*, which can be induced by various mutations, such as frameshift and transposable insertion, can explain the prevalence and rapid evolution of self-compatibility in polyploids.

New: The high frequency of loss-of-function of *SCR* or *SRK*, which can be induced by various mutations, such as frameshift and transposable insertion, can explain the prevalence and rapid evolution of self-compatibility in both polyploids and diploids.

Line 288: What is “transient” about this dominance relationship? Are you suggesting that it was initially dominant, but is now not dominant?

Thank you for the clarification. Initially, when a self-compatible allele is in a low frequency, the dominance relationship is visible. When the allele is fixed as a homozygote, the dominance is not directly observable. We do not mean it is “not dominant” but not observable directly. We explained the content in more detail.

Old: we propose that dominance played a transient role in alleles in diploid species and in each S-locus of allotetraploid species during the evolution of self-compatibility

New: we propose that dominance played a role in heterozygous alleles in diploid species and in each S-locus of allotetraploid species transiently during the evolution of self-compatibility until becoming homozygous.

Line 290: Is it clear whether the mutation under study was inherited from the diploid, or is it a novel mutation in the allopolyploids lineage? The authors state that there are self-compatible populations of

A. lyrata in N. America and Siberia (Line 108), but this point isn't clarified.

> We agree that this question is interesting but is difficult and beyond the scope of this manuscript. Sequence evidence of the order of mutations may be lost by deletions or rearrangements as implied by the study using *Capsella* allopolyploid species (see also our response to the comment on Line 98, "although its relevance in *SCR* dominance regulation is unclear due to the lack of full-length *SCR* sequence in the A genome"). To address this question, population genetic analysis and resequencing combined with functional experiments would be necessary. The data presented by Kolesnikova et al. is consistent with the former (mutation inherited from diploid) although no direct sequence evidence was obtained due to frequent deletions at the S-locus as far as we read it. In this manuscript, we would refrain from speculating on this issue. To reduce extra information, we removed the geographic localities.

Old: although self-compatible individuals of *A. lyrata* have been reported in North America⁵² and Siberia^{51,53,54}.

New: although self-compatible individuals of *A. lyrata* have been reported⁴⁹⁻⁵²

Reviewer #3 (Remarks to the Author):

The authors addressed most of my concerns. I have no further comments.

> Thank you very much for your kind comment.